# Diverse organ-specific localisation of a chemical defence, cyanogenic glycosides, in flowers of eleven species of Proteaceae

Edita Ritmejerytė [1,2,3]*, Berin A. Boughton[2,4], Michael J. Bayly[2], Rebecca E. Miller[1,5]*

**1** School of Ecosystem and Forest Sciences, The University of Melbourne, Richmond, Victoria, Australia, **2** School of BioSciences, The University of Melbourne, Parkville, Victoria, Australia, **3** Australian Institute of Tropical Health and Medicine, James Cook University, Smithfield, Queensland, Australia, **4** Australian National Phenome Centre, Murdoch University, Murdoch, Western Australia, Australia, **5** Royal Botanic Gardens Victoria, South Yarra, Victoria, Australia

* edita.ritmejeryte@jcu.edu.au (ER); rebecca.miller@rbg.vic.gov.au (REM)

**Data Availability Statement:** Data used for this manuscript are available from the following links: Tissue-specific cyanogenic glycoside concentrations and scripts to produce Figs. 1 and

## Abstract

Floral chemical defence strategies remain under-investigated, despite the significance of flowers to plant fitness. We used cyanogenic glycosides (CNglycs)—constitutive secondary metabolites that deter herbivores by releasing hydrogen cyanide, but also play other metabolic roles—to ask whether more apparent floral tissues and those most important for fitness are more defended as predicted by optimal defence theories, and what fine-scale CNglyc localisation reveals about function(s)? Florets of eleven species from the Proteaceae family were dissected to quantitatively compare the distribution of CNglycs within flowers and investigate whether distributions vary with other floral/plant traits. CNglycs were identified and their localisation in florets was revealed by matrix-assisted laser desorption ionisation mass spectrometry imaging (MALDI-MSI). We identified extremely high CNglyc content in floral tissues of several species (>1% CN), highly tissue-specific CNglyc distributions within florets, and substantial interspecific differences in content distributions, not all consistent with optimal defence hypotheses. Four patterns of within-flower CNglyc allocation were identified: greater tissue-specific allocations to (1) anthers, (2) pedicel (and gynophore), (3) pollen presenter, and (4) a more even distribution among tissues with higher content in pistils. Allocation patterns were not correlated with other floral traits (e.g. colour) or taxonomic relatedness. MALDI-MSI identified differential localisation of two tyrosine-derived CNglycs, demonstrating the importance of visualising metabolite localisation, with the diglycoside proteacin in vascular tissues, and monoglycoside dhurrin across floral tissues. High CNglyc content, and diverse, specific within-flower localisations indicate allocations are adaptive, highlighting the importance of further research into the ecological and metabolic roles of floral CNglycs.

## Introduction

Research into the functional significance and evolution of floral traits has mostly emphasised plant-pollinator interactions [1–3], although it is apparent that floral traits evolve in response

2: https://data.mendeley.com/datasets/ttnk9wxtjh/draft?a=5b245a0c-a2d4-4a5b-ad79-1aca4e05c7ab
Signal intensities used to plot box-dot plots for Figs. 3, 4, 6 and 7 DOI: 10.17632/hcd7hr7kbp.1
MALDI-MSI spectra for Fig 3 https://metaspace2020.eu/dataset/2022-11-21_05h28m01s/browser Fig 4 https://metaspace2020.eu/dataset/2022-11-20_02h04m16s/browser Fig 5 https://metaspace2020.eu/dataset/2022-11-20_02h03m16s/browser Fig 6 https://metaspace2020.eu/dataset/2022-11-20_02h07m52s/browser Fig 7 https://metaspace2020.eu/dataset/2022-11-21_05h20m53s/browser.

**Funding:** "This work was supported by an Early Career Researcher grant to Rebecca Miller, and the Holsworth Wildlife Research Endowment & the Ecological Society of Australia, the Albert Shimmins Fund Scholarship and Norma Hilda Schuster Scholarship to Edita Ritmejerytė. Edita Ritmejerytė was a recipient of an Australian Postgraduate Award Scholarship. Rebecca Miller's lectureship received support from the Cybec Foundation. The funders had no role in study design, data collection and analysis, decision to publish, or preparation of the manuscript."

**Competing interests:** The authors have declared that no competing interests exist.

to a diversity of pressures, and florivore pressure can be a strong driver of floral trait evolution [4]. Consequently, floral chemistry research has mostly emphasised pollinator attraction and interactions [5], despite the potential for floral tissues to contain secondary metabolites as chemical defences against florivores. In addition, research into plant chemical defences has in large majority focused on foliar defences, with floral chemical defence strategies understudied [6], despite the sizeable investment of resources in flowers and their obvious importance for plant fitness. Flowers from a wide diversity of plant taxa contain a variety of defence metabolites, including phenolics and flavonoids [7, 8], alkaloids [9–11], iridoid glycosides [12, 13] and cyanogenic glycosides (CNglycs) [14–17]. Moreover, the allocation of resources to floral chemical defence may be substantial given concentrations of these metabolites in floral tissues can be high [e.g. 14, 16–18].

Several plant defence theories make predictions that remain little tested in flowers [6], with the wide range of plant defence theories being primarily tested in vegetative tissues. The Optimal Defence Theory (ODT) predicts that the most vulnerable (soft or exposed) tissues and those predicted to contribute most to plant fitness (valuable) will be defended with higher concentrations of chemical defences [19, 20]. Consistent with ODT, flowers of several species have higher levels of chemical defence than leaves [9, 12, 14, 16, 17], but this pattern is not universal [e.g. 13–15, 21]. The ODT and related Apparency Theory [6, 22] also predict that more conspicuous or accessible flowers or their specific tissues will be more defended.

Floral colour may also affect apparency depending on the colour vision of flower visitors [23, 24]. This is not surprising since colour can be important for attraction of particular pollinators and can also serve as a signal of unpalatability and deterrent for herbivores [25–28]. Consistent with this, red flowers were more likely to be cyanogenic than white flowers across the genus *Hakea* [23], pink and bronze flowers of wild radish (*Raphanus sativus*) had higher glucosinolate content than white or yellow flowers [29, 30], and dark pink and purple Madagascar periwinkle (*Catharanthus roseus*) petals had higher alkaloid content than pink and white cultivars [11].

The ODT and Apparency Theory are also relevant to the distribution of chemical defences within flowers to maximise fitness, predicting, for example, that floral tissues that are more valuable (e.g. those retained for fruit and seed formation) and vulnerable (e.g. exposed or "apparent" to herbivores) to be more defended [22, 27]. Accordingly, variation in the content of defence compounds has been reported within flowers of some species, although it is hard to assess patterns since relatively few studies report content of chemical defences in all floral parts. Some species have higher content of defences in the most valuable and conspicuous floral parts, such as pistillate tissues (stigmas, styles and ovaries) [17, 31–34], but within-flower distributions of defences are not always consistent with ODT predictions [e.g. 31, 35–37]. Taken together, these studies to date suggest that content of defence metabolites within some floral tissues can be high [e.g. 38], very specific, and differ in their allocation among taxa [e.g. 17, 39], thus it is surprising that more studies have not investigated patterns in interspecific variation in chemical defences within floral tissues.

The localisation of defence metabolites at the fine scale within flowers might also reflect that some defence metabolites can have multiple roles, such as nutrient storage. This is the case with CNglycs, a group of nitrogen (N)-based secondary metabolites found in over 3,000 plant species [40]. With currently 112 structures known [41], CNglycs vary in the structures of their amino acid precursor, aglycones and the type and the number of sugar moieties. The primary role of CNglycs is herbivore deterrence by releasing toxic hydrogen cyanide (HCN) upon tissue disruption [42, 43], particularly characteristic of monoglycosides (those containing one sugar moiety). However, CNglycs, especially those containing more than one sugar moiety

(e.g. diglycosides) may play additional roles in plant metabolism, for example, in nutrient storage and transport [40, 44] and other metabolic functions during flowering and fruit development [35, 45–49]. Chemical defences may be costly [45], thus it is assumed that not all tissues will be maximally defended. Cyanogenesis is a constitutive defence that requires N, often a limiting macronutrient, and is therefore a good system for testing defence theories and resource allocation strategies. Moreover, a recently developed robust method to visualise the localisation of these labile glycosides [50] enables the use of Matrix-Assisted Laser Desorption Ionization Mass Spectrometry Imaging (MALDI-MSI) to examine CNglyc localisation within floral tissues and explore the potential function(s) of CNglycs in these tissues [e.g. 17, 18, 39, 51].

Floral cyanogenesis is common in Proteaceae [14, 17, 23, 31, 52, 53], a large flowering plant family of Gondwanan origin consisting of 75 genera [54], with diversity in morphological traits, habitat diversity and wide distribution across all continents in the Southern Hemisphere [54–56]. Here we use Proteaceae taxa that were identified as having cyanogenic inflorescences, from different tribes and subtribes across the family phylogeny [55, 57], different habitats (latitudes), and with different flower colour and flower size, to investigate patterns in floral cyanogenesis. Specifically, we first asked are there differences in CNglyc content between floral tissues, do species differ in their relative floral defence allocation, and is the allocation of CNglycs within flowers consistent with ODT and Apparency Theory, both at the whole plant level and within flowers [e.g., 17, 21]? We hypothesised that flowers would have higher levels of CNglycs than leaves and, based on previous studies of *Grevillea* spp. [31], that pistillate tissues–which may be both more vulnerable (prominent and apparent) and valuable (retained for fruit/seed formation)–will be defended with higher levels of CNglycs. Second, we asked whether interspecific differences in the relative content of CNglycs within flowers correlate with other traits, such as flower colour [e.g., 23] or disturbance response strategy [e.g., 31], or whether they may reflect other factors, such as taxonomic relatedness or latitude (climate)? The latter has been investigated with respect to patterns in herbivory and foliar defences [58–62]. Here we hypothesised that species with orange-red floral colours would have higher levels of floral chemical defence [23, 29, 30] and, consistent with the findings of Lamont [31], that non-sprouting species which rely on regeneration from seed rather than resprouting after a disturbance, would have higher levels of floral defence. Finally, we aimed to identify the specific CNglycs in florets of selected Proteaceae taxa, to determine whether they differed between species and how they are localised, investigating what fine-scale localisation within florets reveals about their potential roles in flowers.

## Materials and methods

### Study species

Species from different tribes and subtribes within Proteaceae, different climatic regions, and with flowers of contrasting colours, morphologies and likely different pollinators were selected for this study. Pollinators for these species are not well documented; where possible we classed them as insect or bird pollinated, based on floral structure, colour and size [23] and anecdotal reports and observations. In total 11 species, each from a different genus, from three tribes and six subtribes were sampled, all from subfamily Grevilleoideae, including two monotypic genera [Table 1; Fig 1; 57]. Three species are known to resprout following disturbance (fire or cyclone), while the rest are known or assumed to rely on regeneration from seed following disturbance (Table 1). Pistil length (used as a measure of relative within-floret prominence/apparency) ranged from 4 to 30.5 mm across species (Table 1).

**Table 1. Details of eleven Proteaceae species in this study (in taxonomic order based on [57]).**

| Species | Tribe | Subtribe | Climate | Habit | Max height (m) | Flower colour | Floret size (mm)[a] | Pistil length (mm) | Disturbance response[b] | Pollinator |
|---|---|---|---|---|---|---|---|---|---|---|
| *Megahertzia amplexicaulis* A.S. George & B.Hyland | Roupaleae Meisn. | - | tropical | tree | 10 | white | 34–39 | 30.5 | non-sprouter[c] | insect |
| *Neorites kevedianus* L.S. Sm. | Roupaleae Meisn. | Roupalinae L.A.S. Johnson & B.G. Briggs | tropical | tree | 30 | white | 4–6 | 4 | non-sprouter[§] | insect |
| *Hollandaea riparia* B. Hyland | Roupaleae Meisn. | Heliciinae L.A.S. Johnson & B.G. Briggs | tropical | shrub/tree | 6 | purple | 18–25 | 12.5 | non-sprouter[c] | insect |
| *Helicia australasica* F. Muell. | Roupaleae Meisn. | Heliciinae L.A.S. Johnson & B.G. Brigg | tropical | shrub/tree | 20 | purple | 15–20 | 20 | non-sprouter[c] | insect |
| *Lomatia myricoides* (C. F.Gaertn.)Domin | Embothrieae Reichb. | Lomatininae L.A.S. Johnson & B.G. Briggs | temperate | shrub/tree | 8 | white | 18–19 | 8 | resprouter | insect |
| *Telopea speciosissima* (Sm.)R.Br. | Embothrieae Reichb. | Embothriinae Endl. | temperate | shrub | 3 | red | 13–59 | 25 | resprouter | bird |
| *Buckinghamia celsissima* F.Muell. | Embothrieae Reichb. | Hakeinae Endl. | tropical | tree | 30 | white | 21–28 | 17.5 | resprouter | insect |
| *Grevillea robusta* A. Cunn. ex R.Br. | Embothrieae Reichb. | Hakeinae Endl. | subtropical | tree | 40 | orange | 29–45 | 25 | non-sprouter (fire-stimulated germination) | bird |
| *Hakea bucculenta* C.A. Gardner | Embothrieae Reichb. | Hakeinae Endl. | temperate | shrub | 4 | red | 18–21 | 19.5 | non-sprouter (fire-stimulated seed release) | bird |
| *Lasjia claudiensis* (C.L. Gross & B.Hyland) P.H. Weston & A.R.Mast | Macadamieae C. Venkata Rao | Macadamiinae L.A. S.Johnson & B.G. Briggs | tropical | tree | 30 | white | ~30 | 15 | non-sprouter[c] | insect |
| *Macadamia tetraphylla* L.A.S.Johnson | Macadamieae C. Venkata Rao | Macadamiinae L.A. S.Johnson & B.G. Briggs | subtropical | tree | 18 | white | 10 | 9.5 | non-sprouter[c] | insect |

[a] Floret size indicates the length from base of pedicel to tip (pollen presenter) of open floret from [63–65].

[b] Disturbance is typically fire in temperate climates and cyclones for species in tropical areas. Data obtained from [63–68] and personal communications from Rigel Jensen, Andrew Ford and Garry Sankowsky and personal observations.

[c] Rarely fireprone (fire killed) considered the ancestral or dominant trait for the genus [66].

## Plant material

Fresh leaves and inflorescences from 1–6 individual plants per species were collected from botanical gardens, private properties and national parks in Australia (collection and herbarium voucher details are in S1 Table). Due to difficulty of obtaining floral material from tall flowering rainforest trees in remote locations, *Megahertzia amplexicaulis* and *Neorites kevedianus* inflorescences could only be obtained from one plant. Plant material was either posted for next day delivery to our laboratory in sealed plastic bags with moist paper towels or kept fresh and transported back to the laboratory on ice in a cold box in sealed plastic bags. Samples were refrigerated until dissected for chemical analysis 1–2 days after field collection [18, 69]. Florets of most species were dissected at immature or partially open flower stages prior to anther dehiscence (S1 Fig), but for four species (*H. australasica*, *H. riparia*, *N. kevedianus* and *M. amplexicaulis*) only fully open florets were available (anthers dehisced) thus anthers could not be separated from flowers of these species.

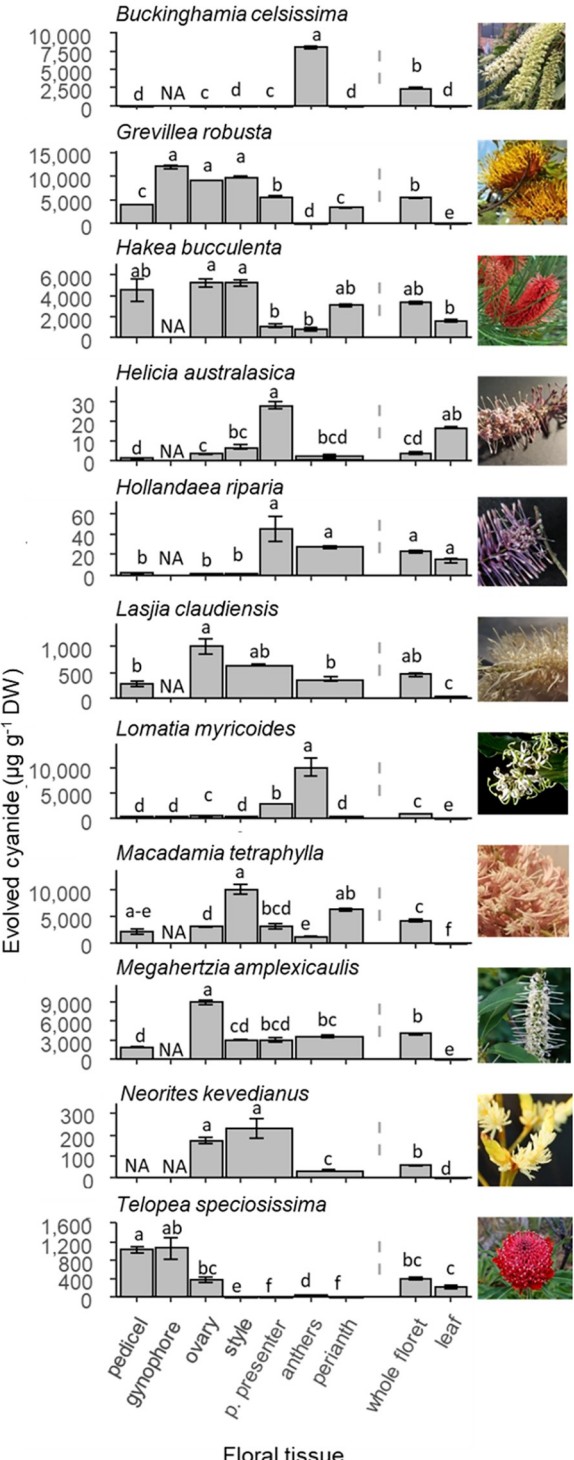

**Fig 1. The content of cyanogenic glycosides measured as evolved cyanide (µg CN g$^{-1}$ DW) in specific floral tissues, whole florets and leaves from eleven Proteaceae species (means ± SE, n = 3–5).** Inflorescence morphology is also shown. Evolved cyanide content differed significantly between floral tissues in all species ($P \leq 0.0001$). Letters (abc) indicate significant differences between floral tissues for the same species, using Tukey HSD family grouping test; means that do not share a letter are significantly different. Interspecific differences in whole flower CNglyc content are in S2 Table. Wide bars indicate the content of two adjacent tissues analysed together. NA–tissue absent for the species. *L. myricoides* data are from [17].

## Chemical analyses

**Quantitative determination of cyanogenesis.** Because the intraspecific patterns in relative within-floret CNglyc allocation were consistent within a species for multiple individuals (S2 Fig) and this consistency has been previously reported for congeneric species [17], 3–5 replicates of tissue consisting of 2–20 florets (depending on floret size) from 1–6 individual trees pooled for a representative bulk sample for each species were assayed for each tissue type. Individual florets from each species were dissected to separately quantify CNglyc content in the pedicel, gynophore (where present), ovary, style, pollen presenter, and perianth (S1 Fig). Anthers, which are sessile and embedded within the perianth, were initially analysed together with the perianth. For two species—*Lasjia claudiensis* and *Neorites kevedianus*—styles and pollen presenters were analysed together due to the smaller size of these tissues in these species, in particular the pollen presenter which is absent in *N. kevedianus* [70], and harder to distinguish from the style in *L. claudiensis*. After initial assays showed high relative content of CNglycs in perianth tissue which included anthers, fresh samples of 6 species (*Buckinghamia celsissima*, *Grevillea robusta*, *Hakea bucculenta*, *Lomatia myricoides*, *Macadamia tetraphylla* and *Telopea speciosissima*) were available to quantify CNglycs in pollen (anthers) and perianth separately [for *L. myricoides* see 17]. For these six species the anthers were removed from the perianth prior to dehiscence for separate analysis. Mature leaves were sampled and analysed for all trees from which florets were sampled.

The content of CNglycs was measured as evolved cyanide following the method for fresh tissue described in [71]. Briefly, between 2 and 100 mg FW of tissue was added to 0.4 mL of 0.1 M citrate buffer-HCl (pH 5.5) with a non-specific β-glycosidase, emulsin (1.04 units $mL^{-1}$), and was subjected to three freeze-thaw cycles to lyse the tissue. The mass range for tissues analysed reflects the different sizes of dissected floral tissues. Following incubation for 16–20 h at 37˚C cyanide trapped in a 1 M NaOH well was quantified using a colorimetric assay and photometric microplate reader with incubator (Multiskan GO, Thermo Scientific, Finland) [72]. Assayed tissues were rinsed and oven dried to enable determination of CNglyc content on a dry weight basis (μg CN $g^{-1}$ DW). The content of CNglycs in the whole floret was determined from the tissue masses and evolved cyanide content of dissected tissues. Assays of whole floret tissues conducted with and without the addition of emulsin were used to confirm cyanogenic capacity (i.e. presence of both cyanogenic glycoside and β-glycosidase).

**Extraction and identification of cyanogenic glycosides.** To enable MALDI-MSI visualisation of the fine-scale localisation of CNglycs, it was first necessary to determine the specific CNglycs in floral tissues of the study species. CNglycs were extracted from whole florets of *G. robusta*, *H. bucculenta*, *T. speciosissima*, *M. tetraphylla* and *N. kevedianus* using one of two standard methods. For all species except *T. speciosissima* and *Polyscias australiana* (F.Muell.) (Araliaceae), which was used as a reference of known CNglyc composition, CNglycs were extracted using the hot methanol procedure [17, 18, 71]. Extracts of *T. speciosissima* and *P. australiana* tissue followed the procedure described in [73] where freeze-dried tissue (10–20 g) was de-fatted by a minimum of three extractions with petroleum ether (solvent:tissue, 10:1 v/w), filtered (Whatman® 541 filter paper, Whatman Asia Pacific, San Centre, Singapore), and twice extracted with cold methanol, and filtered. The filtrate volume was reduced by rotary evaporation (40˚C), and an equivalent volume of $CHCl_3$ added, with sufficient $H_2O$ to facilitate phase separation. The MeOH phase was concentrated *in vacuo* and freeze dried. Because some Proteaceae are known to contain tyrosine-derived CNglycs [53], floral extracts of the six species were analysed alongside an extract from *P. australiana* foliage as a standard because it is known to contain two tyrosine-derived CNglycs, dhurrin and proteacin [73].

The extracts were also analysed by LC-MS. Cooled MeOH extracts of *G. robusta*, *H. bucculenta*, *T. speciosissima*, *M. tetraphylla*, *N. kevedianus* and *P. australiana* were diluted ~5 times in $H_2O$ and filtered through a syringe filter (0.45 μm, Merck Millipore) for liquid chromatography tandem mass spectrometry (LC-MS/MS) using an Agilent 1200 HPLC system coupled to an Agilent 6520 Series QTOF-MS (Agilent Technologies, Mulgrave, VIC, Australia) and A Zorbax SB—C18 column (Agilent; 1.8 μm, 2.1 × 50 mm) as described in [17]. Extracted ion chromatograms for specific $[M + Na]^+$ adduct ions were used to locate peaks, then their MS, MS/MS and RT were used to identify compounds. The LC–MS data were analysed using Agilent MassHunter Qualitative Analysis B.07.00 (Agilent Technologies).

**Maldi mass spectrometry imaging.** Species with the highest floral CNglyc content, and different distributions of CNglycs within tissues (based on dissection assays) were used for CNglyc visualisation by MALDI-MSI. Specifically, sections of *G. robusta*, *M. tetraphylla*, *T. speciosissima* and *H. bucculenta* florets were imaged. In addition, young fruits of *N. kevedianus* were opportunistically imaged as one of the species with high CNglycs in the ovary, and rachis cross sections of *H. bucculenta*, a species with high CNglycs in leaves and inflorescences, were imaged to examine the potential for CNglyc transport. Sections were imaged using the methods and instrument settings identical to those described in [17, 18]. The area selected for imaging was defined using flexImaging software (v4.1, Bruker, Bremen, Germany). Data were imported into SCiLS lab software (version 2017a) where the respective Regions of Interest (ROIs) were selected, signal intensities of each region were exported and box-dot plots were generated.

## Light microscopy

To visualise the tissue layers and inform interpretation of the MALDI-MS images of *H. bucculenta* rachis cross sections, samples of rachis (~ 5 × 5 mm) were collected for anatomical imaging. These samples were immersed in fixative composed of 2.5% (v/v) glutaraldehyde and 4% (v/v) paraformaldehyde in phosphate-buffered saline (PBS) solution (Gibco® PBS tablets) for 3 h, then washed three times in PBS for 10 min. Fixed tissues were dehydrated in an ethanol series (10, 20, 40, 60, 80, 100%) for at least two h at each stage, and in 100% EtOH overnight. London Resin White (LRW) was infiltrated in series (25, 50, 75, 100, 100, 100%) for at least one h at each stage. The tissues were placed in gelatine capsules, submerged in LRW and polymerised overnight at 60˚C. Embedded tissues were cross sectioned (0.5 μm thick) with Leica Ultracut R microtome, and stained with 1% Toluidine Blue in 1% sodium tetraborate. Sections were visualised with Leica DM6000 compound microscope, and images captured with Leica 450C colour camera (acquisition software used was MetaMorph; Molecular Devices).

## Statistical analyses

The content of CNglycs in different tissues within and between species were first tested for normality and homogeneity using Shapiro-Wilk's and Levene's tests, respectively. Data not meeting these assumptions were log transformed. Transformed data were compared using Tukey HSD *post-hoc* test with significance at $P \leq 0.05$ using Minitab (version 17). Linear model function *lm* in 'stats' package for R studio [74] was used to predict effects of leaf cyanide content, flower colour (white, pink-purple, orange-red), species climatic zone (temperate, subtropical and tropical), pistil length (<5, 5–10, 10–20, 20–30 mm), regeneration strategy (resprouter or non-sprouter), tribe (Embothrieae, Macadamieae, Roupaleae) on whole flower and tissue specific cyanide content, followed by Tukey-Kramer *post-hoc* analysis to account for uneven sample size. To investigate patterns and potential species groupings based on tissue-specific floral defence strategies and CNglyc allocation patterns, relative CNglyc content was

used, where the tissue with the highest content was scored 1 for each species, and content in other tissues was expressed relative to that value. Relative CNglyc content was visualised as heatmap and grouped using *hclust* model function for Hierarchical cluster analysis in 'ggdendro' package [75] to assess patterns in CNglycs allocation against other traits/factors. Data were visualised in the R Studio [74], using the 'ggplot2', and 'plyr' packages [76, 77].

## Results

### Interspecific differences in whole floret and leaf cyanogenic glycoside content

Whole floret CNglyc content tended to fall into 3 groups–high, medium and low (Fig 1). High floral CNglyc content (2,359–5,578 µg CN $g^{-1}$ DW) was measured in *G. robusta*, *M. tetraphylla*, *H. bucculenta*, *Megahertzia amplexicaulis*, and *B. celsissima*; medium CNglyc content (385–668 µg CN $g^{-1}$ DW) was measured in *L. myricoides*, *L. claudiensis*, and *T. speciosissima*; and florets of *Helicia australasica*, *Hollandaea riparia* and *N. kevedianus* had low content (3.6–58.2 µg CN $g^{-1}$ DW) (Fig 1; S2 Table).

Foliar CNglyc content also differed significantly between species (0.1–1,644 µg CN $g^{-1}$ DW), and the general linear model failed to detect a relationship between floral and foliar CNglyc content across species [log(whole flower CNglyc content)~log(foliar CNglyc content) +(1|species), $R^2$ = 0.02, *P* = 0.29]. Floral CNglyc content was higher than foliar content (seven species) while in four species (*H. bucculenta*, *H. riparia*, *T. speciosissima*, and *H. australasica*) they were similar (Fig 1). *H. bucculenta* had the highest foliar CNglyc content (1,643.8 µg CN $g^{-1}$ DW) and among the highest floral content (3,359.3 µg CN $g^{-1}$ DW), whereas *G. robusta* with the highest floral CNglyc content had only 6.7 µg CN $g^{-1}$ DW in leaves. Leaves of five species had very low CNglyc content (7–17 µg CN $g^{-1}$ DW), and in *B. celsissima*, *L. myricoides* and *N. kevedianus* content was negligible ($<$ 2.4 µg CN $g^{-1}$ DW). Floral CNglyc content ranged widely (23–5,578 µg CN $g^{-1}$ DW) in species with weakly cyanogenic foliage (Fig 1; S2 Table).

### Interspecific differences in the content and allocation of CNglycs within florets

The quantification of CNglycs in floral tissues revealed marked differences in the content distributions of CNglycs within flowers of different genera and species (Figs 1 and 2). In some species, CNglycs were found in all floral tissues, whereas in other species higher CNglyc content was restricted to one or two specific tissues with low to negligible levels elsewhere. Seven species, had the highest relative CNglyc content in at least one pistillate tissue (ovary, style or pollen presenter), and in three species, the highest relative CNglyc content was in non-pistillate tissues. Among species with more tissue-specific distributions, *B. celsissima* had the highest CNglyc content in the anthers (8,089 µg CN $g^{-1}$ DW), with low to negligible content in other floral tissues (0.1–18 µg CN $g^{-1}$ DW), *L. myricoides* as reported in Ritmejerytė, Boughton [17] had the highest CNglyc content in anthers (and specialised cells on pollen presenter), and *T. speciosissima* had highest CNglyc content in pedicels and gynophores (Figs 1 and 2). Among the six species for which anthers were analysed separately, there was no consistent pattern in levels of pollen CNglyc content with respect to other tissues. In contrast to the high pollen-specific content of CNglycs in *B. celsissima* and in *L. myricoides*, for the other four species, *G. robusta*, *H. bucculenta*, *M. tetraphylla* and *T. speciosissima*, anther CNglyc content was among the lowest of all floral tissues (Fig 1).

CNglyc content also varied significantly within pistils of several species with no consistent pattern. For example, in *M. amplexicaulis* CNglyc content in the ovary was greater than the

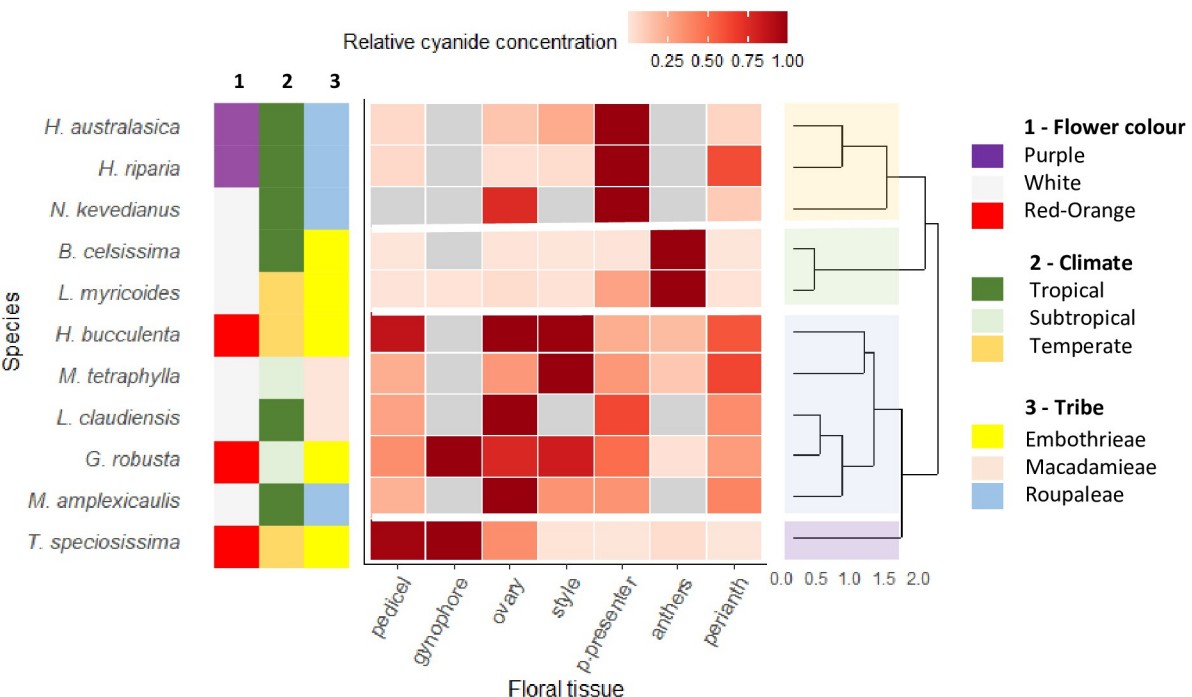

**Fig 2. Hierarchical cluster analysis and heatmap of relative cyanide content in eleven Proteaceae species, with respect to three variables: Flower colour, climate (latitude) and tribe.** For *L. claudiensis* and *N. kevedianus* "pollen presenter" the proportion of total evolved cyanide value represents the combined content of style and pollen presenter assayed together. Anthers were analysed with perianth for *H. australasica, H. riparia, N. kevedianus, L. claudiensis* and *M. amplexicaulis*. Gynophores were only present in *L. myricoides, T. speciosissima* and *G. robusta* florets. *N. kevedianus* florets do not have pedicels. Grey squares indicate no value. Four major groups of CNglyc allocation patterns within flowers were identified indicated by different colour shading in the dendrogram.

style and pollen presenter, in *M. tetraphylla* style content was greater than ovary and pollen presenter levels, and in *H. riparia* and *H. australasica* pollen presenter content was greater than those in the ovary and style (Fig 1).

Cluster analysis based on the relative content of CNglycs in each floral tissue (content expressed relative to the tissue with highest content for each species) (Fig 2) identified four patterns in the relative CNglyc allocation within florets. Three patterns were characterised by a more specific CNglyc allocation to particular floral tissues, and the fourth and largest group of five species was characterised by a more even distribution of CNglycs across all floral tissues and higher content in pistillate tissues. This latter group comprised *H. bucculenta, M. tetraphylla, L. claudiensis, G. robusta* and *M. amplexicaulis*. Dominance of anther CNglyc levels grouped *B. celcissima* and *L. myricoides* together, and greater relative pollen presenter CNglycs grouped *H. australisca, H. riparia* and *N. kevedianus*, whilst *T. speciossisima*, with its unique pedicel and gynophore dominant pattern, formed its own group.

## Identification of cyanogenic glycosides in Proteaceae flowers

The tyrosine-derived cyanogenic monoglycoside dhurrin and related diglycoside proteacin were identified as the cyanogenic glycosides in *G. robusta, H. bucculenta, T. speciosissima, N. kevedianus* and *M. tetraphylla* using a combination of analytical techniques (TLC, LC-MS and LC-MS/MS) and a *Polyscias australiana* leaf MeOH extract known to contain both dhurrin and proteacin [73] as a standard.

The quantification of CNglycs in TLC plate scrapings showed that floral (and foliar in some species) extracts of all six species had two regions with cyanogenic activity similar to the standard foliar extract of *P. australiana*. The first cyanogenic band was unresolved and remained near the origin ($R_f = 0.1–0.2$), similar to proteacin in the *P. australiana* foliar extract [73]. Proteacin was the major cyanogen in *N. kevedianus* and *T. speciosissima* floral extracts, accounting for 99 and 96% of the total floral CNglycs, respectively. The second cyanogenic band resolved at $R_f = 0.6$ (except for *N. kevedianus* $R_f = 0.8$) and was consistent with the dhurrin standard from *P. australiana* foliar extract. Dhurrin was the major cyanogen in floral extracts of *G. robusta* (89% total floral CNglycs), *H. bucculenta* (78%) and the foliar extract of *M. tetraphylla* (97.2%).

To confirm CNglyc identities, floral and foliar methanol extracts were analysed by LC–MS, which showed that proteacin in the *P. australiana* extract eluted as a single peak at a retention time of 4.3 min with *m/z* 496.1435 ($\Delta m = 1.9$ mDa) (S3 Fig). In floral extracts of *G. robusta*, *H. bucculenta* and *T. speciosissima*, proteacin eluted as a single peak at retention times between 4.2–4.4 min with *m/z* 496.14 ($\Delta m = 2$ mDa), and for *N. kevedianus* it eluted at 4.8 min, with *m/z* 496.1424 ($\Delta m = 0.3$ mDa). The LC-MS/MS ion fragmentation of proteacin provided the same diagnostic fragments (corresponding to the loss of 27 Da, HCN) at *m/z* 469.13 for *P. australiana*, *G. robusta*, *H. bucculenta*, *N. kevedianus* and *T. speciosissima* extracts. Fragments corresponding to a further loss of glucose, i.e. $[M—Glc–HCN + Na]^+$ at *m/z* 307.08; and fragments corresponding to $[Glc–H_2O + Na]^+$ at *m/z* 185.04 were also detected in the extracts of these species. Proteacin was not detected in the foliar extract of *M. tetraphylla*.

Extracted Ion Chromatograms (EIC) from LC-MS showed that dhurrin in the *P. australiana* extract eluted as a single peak at retention time of 5.9 min with *m/z* 334.0894 ($\Delta m = 1$ mDa), and consistent with the presence of dhurrin in extracts from *G. robusta*, *H. bucculenta*, *M. tetraphylla*, *N. kevedianus* and *T. speciosissima*, a peak with *m/z* 334.09 (S3 Fig) and mass error $<4$ mDa eluted at retention times between 5.9 and 6.0 min. Diagnostic fragment ions of dhurrin were confirmed in Proteaceae extracts against the *P. australiana* extract as a standard for all species except *N. kevedianus* and *T. speciosissima* where the signal for dhurrin was too low (accounting only for 1–4% of total CN). Specifically, ions corresponding to the loss of 27 Da (HCN) were at *m/z* 307.08 for *P. australiana*, *G. robusta* and *H. bucculenta* extracts. $[Glc–H_2O + Na]^+$ fragment ions were detected at *m/z* 185.04 in *P. australiana*, *G, robusta*, *H. bucculenta* and *M. tetraphylla* extracts.

## Detection of cyanogenic glycosides in Proteaceae flowers using MALDI-MSI

In addition to the within-floret variation in CNglyc content characterised using floral dissections (Fig 1), MALDI-MSI identified distinct and tissue-specific distributions of dhurrin and proteacin across floral tissues of *G. robusta*, *N. kevedianus*, *M. tetraphylla*, *T. speciosissima* and rachises of *H. bucculenta* (Figs 3–7). For most floral tissues, dhurrin was detected as two salt adducts: $[M + Na]^+$ (*m/z* 334.0896, $\Delta m = 0.3$ mDa) and $[M + K]^+$ (*m/z* 350.0635, $\Delta m = 0.6$ mDa); with proteacin also detected as the same adducts: $[M + Na]^+$ (*m/z* 496.1421, $\Delta m = 0.8$ mDa) and $[M + K]^+$ (*m/z* 512.1164, $\Delta m = 0.2$ mDa). The ratio of the dhurrin to proteacin varied between floral tissues and between species (S4 Table).

In *G. robusta* florets, dhurrin and proteacin were distributed across all floral tissues and co-localised, although epidermal layers of the pistil appeared to contain more dhurrin than proteacin, which was restricted to vascular tissues of a senescent floret, and, consistent with the TLC results, proteacin had overall lower signal intensity in floral tissues than dhurrin (Fig 3E and 3F). Relative signal intensities (assuming similar ionisation efficiencies) determined that

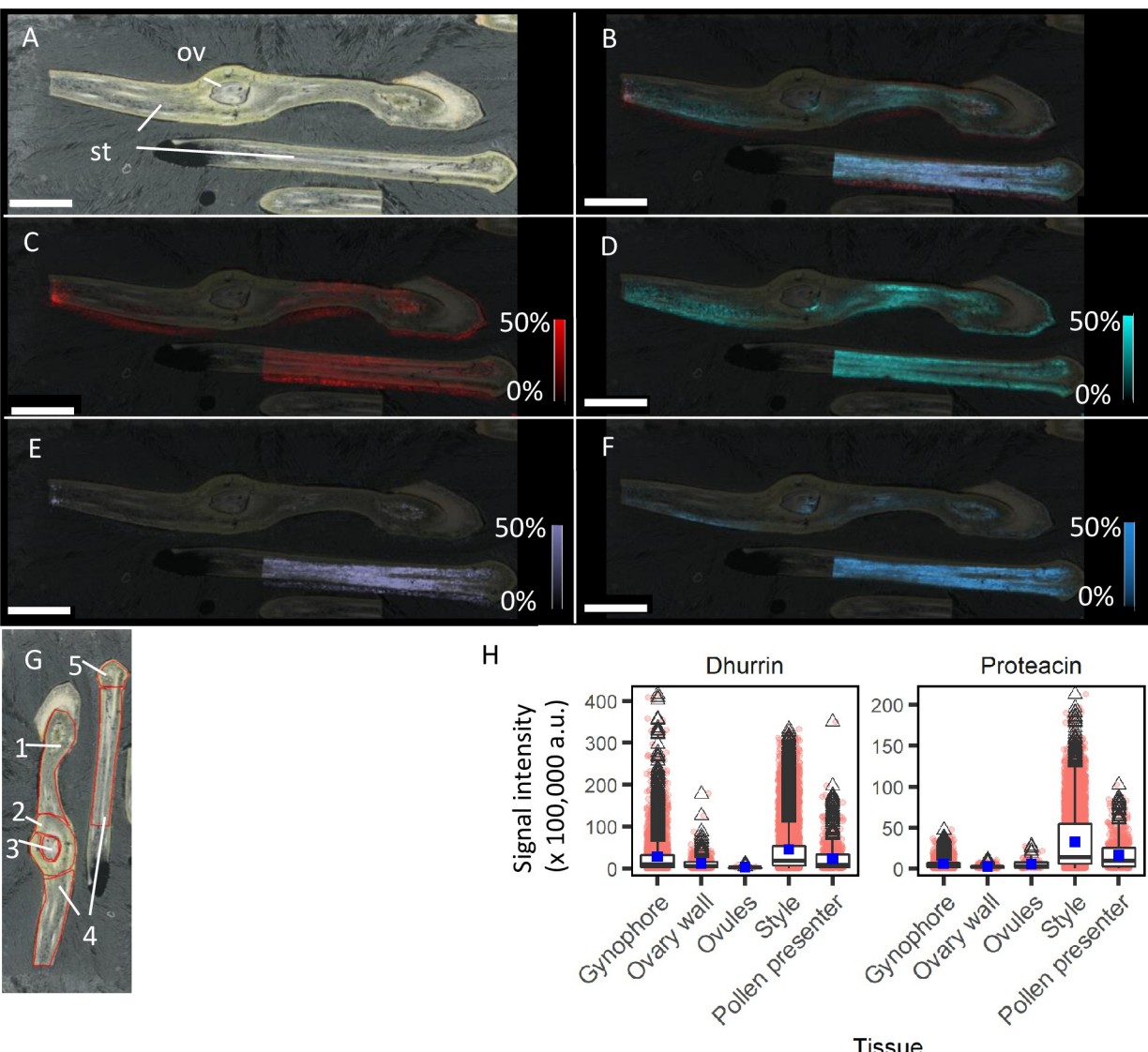

**Fig 3. MALDI-MS images of a longitudinal section of a senescent floret of *Grevillea robusta*.** (A) Optical image of section (35 μm thick), where st is style, ov–ovules. (B) Merged positive ion MALDI-MS image showing dhurrin and proteacin, (C) dhurrin $[M + Na]^+$ (*m/z* 334.0896, Δm 0.3 mDa), (D) dhurrin $[M + K]^+$ (*m/z* 350.0635, Δm 0.6 mDa), (E) proteacin $[M + Na]^+$ (*m/z* 496.1421, Δm 0.8 mDa) and (F) proteacin $[M + K]^+$ (*m/z* 512.1164, Δm 0.2 mDa). All MALDI-MS images were obtained using 50 × 50 μm array. White scale bars indicate 2 mm. Coloured intensity scales represents 0–50% TIC normalised signal intensity. (G) Regions of Interest (ROIs) selected for cyanogenic glycoside signal intensities comparison analysis–(1) gynophore, (2) ovary wall, (3) ovules, (4) style, (5) pollen presenter. (H) Signal intensities of dhurrin ($[M + K]^+$ (*m/z* 350.0635), and $[M + Na]^+$ (*m/z* 334.0896) combined), and proteacin ($[M + K]^+$ (*m/z* 512.1164) and $[M + Na]^+$ (*m/z* 496.1421) combined) for respective ROIs; the box plot represents the entire signal intensity delimited by the 25th and 75th percentiles, with error bars denoting the 10th and 90th percentiles. Values outside those ranges are represented by open triangles, the mean (dark blue square) and median (solid line). Pink dots represent signal intensity in individual pixel. Dhurrin to proteacin ratios in tissues are provided in S4 Table.

dhurrin was more abundant than proteacin in all tissues (58–83%) except ovules at the post-anthesis flower stage (proteacin 66%) (Fig 4H). Conversely, little co-localisation between dhurrin and proteacin was found in partially open florets of *M. tetraphylla* (Fig 4) in which dhurrin was localised to the epidermal layers of the style in particular, and proteacin was detected throughout floral tissues including vasculature. Like in *G. robusta*, however, proteacin was more abundant in the *M. tetraphylla* ovary (74%), whilst dhurrin was more abundant in the

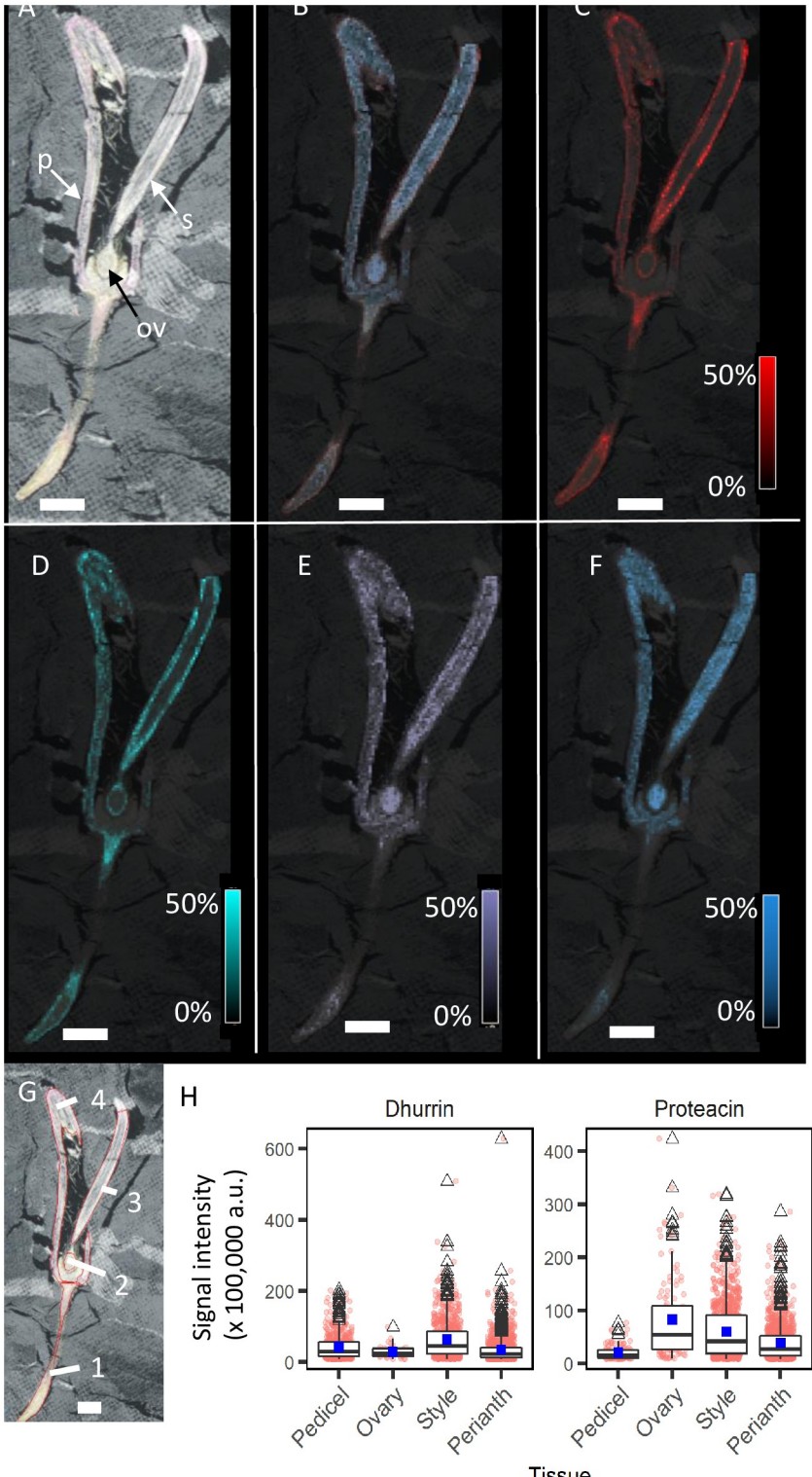

**Fig 4. MALDI-MS images of longitudinal section of *Macadamia tetraphylla* floret at the partially open stage.** (A) Optical image of floret longitudinal-section (35 μm thick), where p is perianth, s–style and ov–ovules. (B) Merged positive ion MALDI-MS image showing dhurrin and proteacin, (C) dhurrin $[M + Na]^+$ (*m/z* 334.0896, Δm 0.3 mDa), (D) dhurrin $[M + K]^+$ (*m/z* 350.0635, Δm 0.6 mDa), (E) proteacin $[M + Na]^+$ (*m/z* 496.1421, Δm 0.8 mDa) and (F) proteacin $[M + K]^+$ (*m/z* 512.1164, Δm 0.2 mDa). All MALDI-MS images were obtained using 50 × 50 μm array. White scale bars indicate 1 mm. Coloured intensity scales represent 0–50% TIC normalised signal intensity. (G)

Regions of Interest (ROIs) selected for cyanogenic glycoside signal intensities comparison analysis–(1) pedicel, (2) ovary, (3) style, (4) perianth. (H) Signal intensities of dhurrin ([M + K]$^+$ ($m/z$ 350.0635), and [M + Na]$^+$ ($m/z$ 334.0896) combined), and proteacin ([M + K]$^+$ ($m/z$ 512.1164) and [M + Na]$^+$ ($m/z$ 496.1421) combined) for respective ROIs, the box plot represents the entire signal intensity delimited by the 25$^{th}$ and 75$^{th}$ percentiles, with error bars denoting the 10$^{th}$ and 90$^{th}$ percentiles. Values outside those ranges are represented by open triangles, the mean (dark blue square) and median (solid line). Pink dots represent signal intensity in individual pixel. Dhurrin to proteacin ratios in tissues are provided in S4 Table.

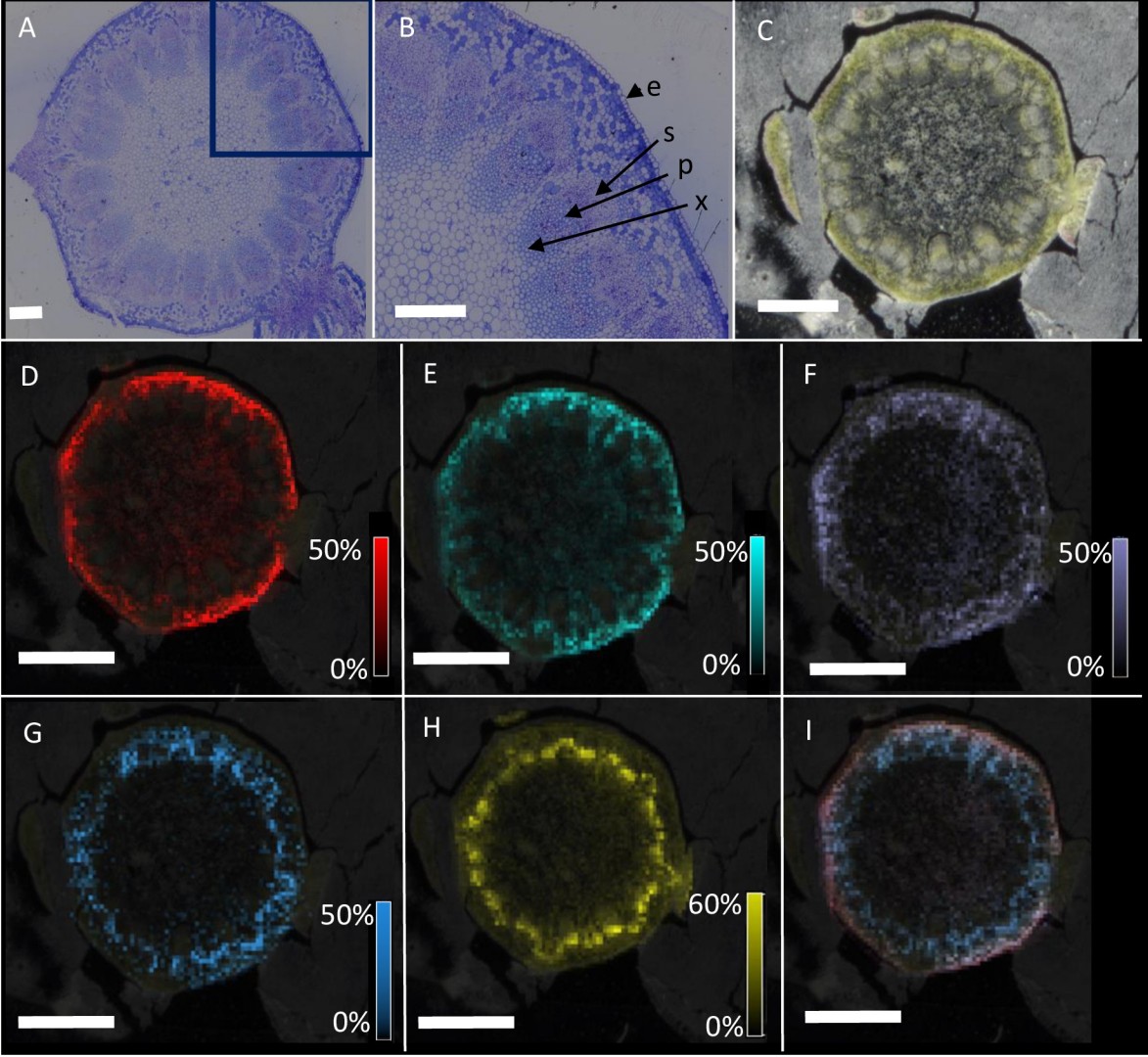

**Fig 5. Light microscopy and MALDI-MS images of transverse section of *Hakea bucculenta* rachises.** (A) Light microscopy image of rachis transverse section. (B) inset of rachis vasculature (box from A), where e is epidermis, x–xylem, p–phloem, s–sclerenchyma. (C) Optical image of rachis transverse-section imaged by MALDI-MS (35 μm thick). (D) dhurrin [M + Na]$^+$ ($m/z$ 334.0896, Δm 0.3 mDa), (E) dhurrin [M + K]$^+$ ($m/z$ 350.0635, Δm 0.6 mDa), (F) proteacin [M + Na]$^+$ ($m/z$ 496.1421, Δm 0.8 mDa), (G) proteacin [M + K]$^+$ ($m/z$ 512.1164, Δm 0.2 mDa), (H) hexose [M + K]$^+$ ($m/z$ 381.0792, Δm 0.5 mDa), and (I) Merged positive ion MALDI-MS image of rachis transverse-section showing dhurrin and proteacin. All MALDI-MS images were obtained using 50 × 50 μm array. White scale bars indicate 200 μm for A-B, and 1 mm for C-I. Coloured intensity scales represent 0–50% TIC normalised signal intensity (0–60% for hexose).

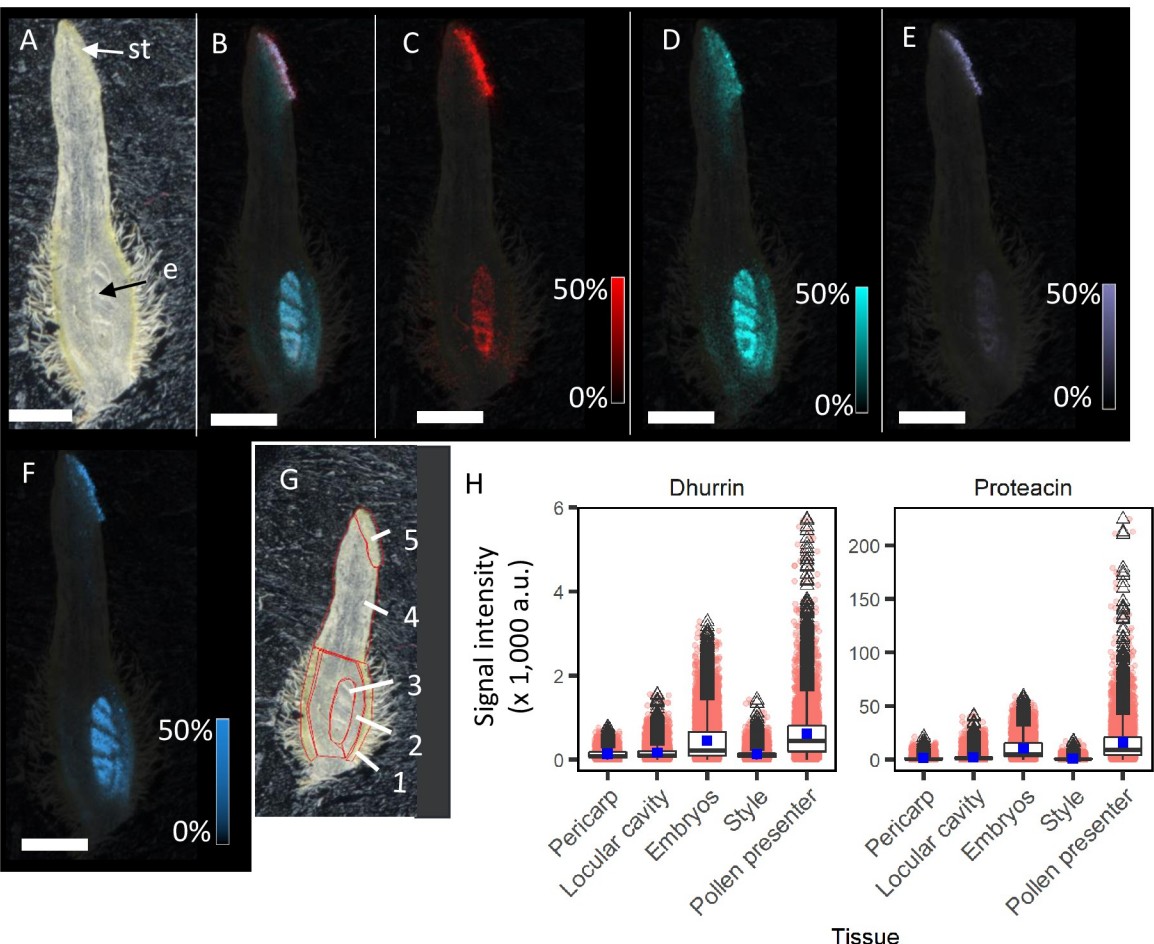

**Fig 6. MALDI-MS images of longitudinal section of *Neorites kevedianus* fertilised floret.** (A) Optical image of young fruit longitudinal-section (35 μm thick), where st is stigmatic surface, e–embryos. (B) Merged positive ion MALDI-MS image showing dhurrin and proteacin, (C) dhurrin [M + Na]$^+$ (*m/z* 334.0896, Δm 0.3 mDa), (D) dhurrin [M + K]$^+$ (*m/z* 350.0635, Δm 0.6 mDa), (E) proteacin [M + Na]$^+$ (*m/z* 496.1421 Δm 0.8 mDa) and (F) proteacin [M + K]$^+$ (*m/z* 512.1164, Δm 0.2 mDa). All MALDI-MS images were obtained using 30 × 30 μm array. White scale bars indicate 1 mm. Coloured intensity scales represent 0–50% TIC normalised signal intensity. (G) Regions of Interest (ROIs) selected for cyanogenic glycoside signal intensities comparison analysis–(1) pericarp, (2) locular cavity, (3) embryos, (4) style, (5) pollen presenter. (H) Signal intensities of dhurrin ([M + K]$^+$ (*m/z* 350.0635), and [M + Na]$^+$ (*m/z* 334.0896) combined), and proteacin ([M + K]$^+$ (*m/z* 512.1164) and [M + Na]$^+$ (*m/z* 496.1421) combined) for respective ROIs, the box plot represents the entire signal intensity delimited by the 25$^{th}$ and 75$^{th}$ percentiles, with error bars denoting the 10$^{th}$ and 90$^{th}$ percentiles. Values outside those ranges are represented by open triangles, the mean (dark blue square) and median (solid line). Pink dots represent signal intensity in individual pixel. Dhurrin to proteacin ratios in tissues are provided in S4 Table.

pedicel (67%), and almost equal proportions the two CNglycs occurred in the styles and perianth (Fig 4H). Visualisation of CNglycs in cross sections of *H. bucculenta* rachises to investigate whether CNglycs may be transported around inflorescences strongly suggest that the diglycoside proteacin co-localised with hexose sugars in phloem whereas dhurrin was localised to the epidermis and cortex (Fig 5).

Although both dhurrin and proteacin were co-localised in young fruit of *N. kevedianus*, both were specifically confined to the stigmatic surface and developing seed embryos (Fig 6). This specificity was not revealed by the quantitative assays since pollen presenters and styles were analysed together (Fig 1), and across all tissues proteacin was the major CNglyc in *N. kevedianus* (86–96%; Fig 5H, S4 Table). Similarly, a very specific localisation of proteacin to

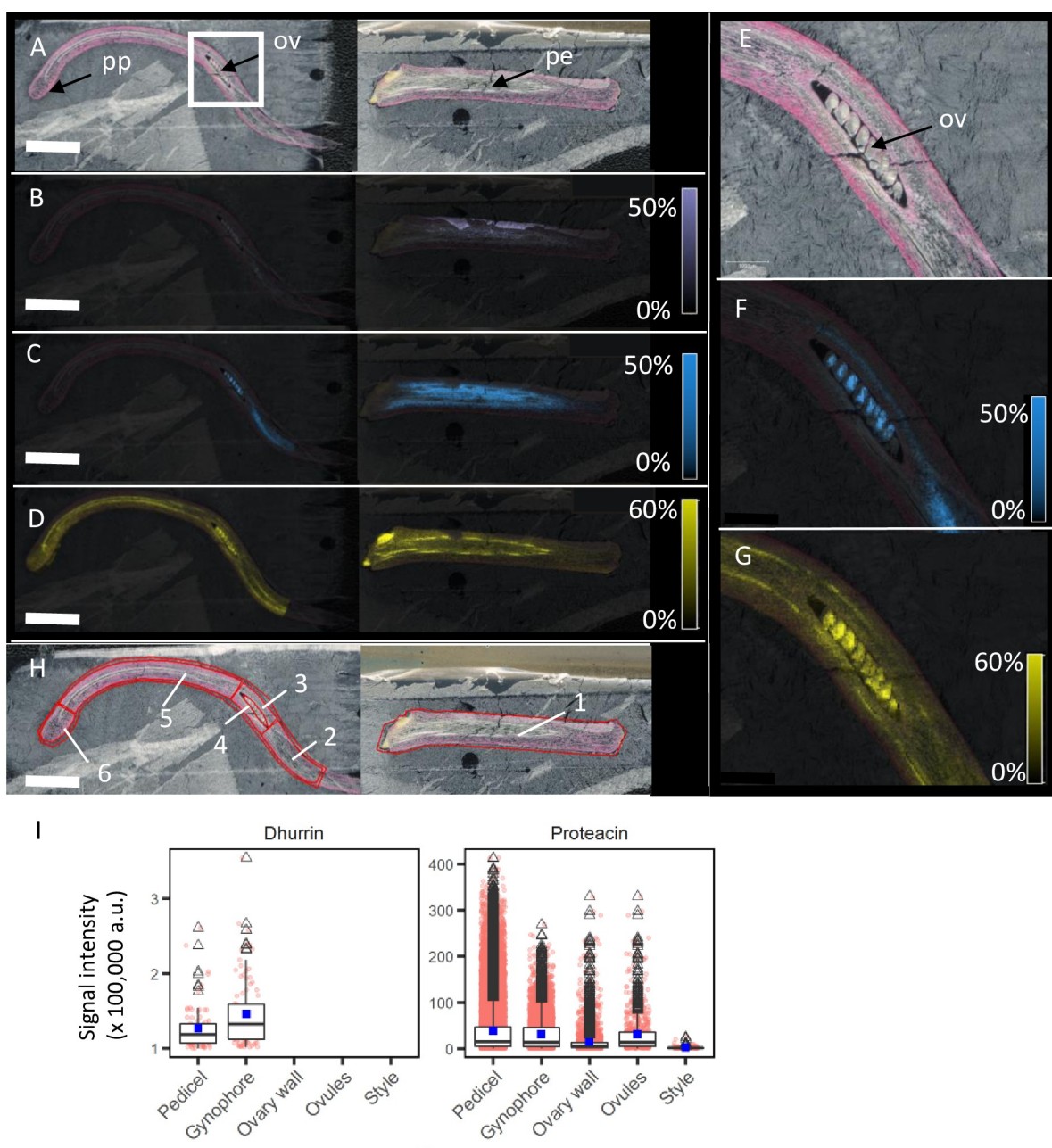

**Fig 7. MALDI-MS images of longitudinal sections of a *Telopea speciosissima* floret without perianth showing the pistil (left) and pedicel (right).** (A) Optical image of floret longitudinal-section (35 μm thick), where pp is pollen presenter, ov–ovules, pe–pedicel. (B) proteacin [M + Na]$^+$ (*m/z* 496.1421, Δm 0.8 mDa), (C) proteacin [M + K]$^+$ (*m/z* 512.1164, Δm 0.2 mDa) and (D) hexose [M + K]$^+$ (*m/z* 381.0792, Δm 0.5 mDa). (E) Inset of *T. speciosissima* ovary (white square from figure A), (F) proteacin [M + K]$^+$ (*m/z* 512.1164, Δm 0.2 mDa), (G) hexose [M + K]$^+$ (*m/z* 381.0792, Δm 0.5 mDa). All MALDI-MS images were obtained using 50 × 50 μm array. White scale bars indicate 5 mm. Coloured intensity scales represent 0–50% RMS normalised signal intensity (0–60% for hexose). (H) Regions of Interest (ROIs) selected for cyanogenic glycoside signal intensities comparison analysis–(1) pedicel, (2) gynophore, (3) ovary wall, (4) ovules, (5) style, (6) pollen presenter. (I) Signal intensities of dhurrin ([M + K]$^+$ (*m/z* 350.0635)), and proteacin ([M + K]$^+$ (*m/z* 512.1164) and [M + Na]$^+$ (*m/z* 496.1421) combined) for respective ROIs, the box plot represents the entire signal intensity delimited by the 25th and 75th percentiles, with error bars denoting the 10th and 90th percentiles. Values outside those ranges are represented by open triangles, the mean (dark blue square) and median (solid line). Pink dots represent signal intensity in individual pixel. Dhurrin to proteacin ratios in tissues are provided in S4 Table.

the ovules in open florets of *T. speciosissima* was identified (Fig 7), which overlapped with localisation of hexose sugars, and consistent with TLC and LC-MS results, proteacin was the major glycoside (accounting for 96–100% of total floral CNglycs) and dhurrin was only detected in pedicels (3%) and gynophores (4%) (Fig 7I; S4 Table).

## Interspecific differences in floral cyanogenesis: Taxonomy and other plant and floral traits

**Taxonomic relationships and floral cyanogenesis.**   We investigated if the pattern of CNglyc allocation within flowers of different species clustered in relation to their position in the Proteaceae phylogeny characterised by Weston and Barker (2006) and Carpenter (2012) [55, 57] and found no consistent pattern (S4A Fig). CNglyc allocation patterns showed that across the eleven species studied, Macadamieae had significantly higher relative content in styles than the other two tribes. Pollen presenters had the highest relative CNglyc content in Roupaleae, compared to the other two tribes. For the remaining floral tissues, there were no consistent allocation patterns amongst these three tribes (Fig 2).

**Flower colour, putative pollination system, floret structure and floral cyanogenesis.** Some differences in floral cyanogenic traits were identified between species with flowers of different colours in comparisons of species with white/cream, orange/red and pink/purple florets, despite considerable variation in content among species with flowers of the same colour (Fig 1). Tissue-specific relative CNglyc content was similar between white/cream and purple flowered species and generally lower than those in orange/red flowered species (S4B Fig), with an exception for pollen presenters that had the higher relative CNglyc content in purple flowers when compared to orange/red and white flowers. By contrast, species with white/cream flowers, most of which are known to be insect pollinated (Table 1) had significantly higher relative CNglyc content in anthers than bird-pollinated orange/red flowered species (Table 1). On the whole, however, species with similar flower colours (and pollinators) did not cluster together based on the patterns in relative CNglyc content within florets (Fig 2; S4B Fig). There were no patterns with respect to pistil length, which was used as a measure of prominence (apparency), and CNglyc content within floral tissues (S4C Fig).

**Latitude, disturbance response strategies and floral cyanogenesis.**   We investigated if there were any latitudinal trends in levels of floral CNglycs among species. Whole floret CNglyc content varied widely among both tropical rainforest and temperate species, while the two subtropical species (*M. tetraphylla* and *G. robusta*) had high floral CNglyc content (Fig 1) and were characterised by significantly higher relative tissue-specific CNglyc content for all floral tissues except anthers, with higher anther CNglyc levels measured in tropical and temperate species (S4D Fig). Similarly, there was no consistent difference in the pattern of CNglyc allocation within florets between tropical rainforest and temperate forest species (Fig 2).

To test whether non-sprouter species, which rely on regeneration from seed following disturbance, invest in higher levels of floral defence than species which rely on resprouting from basal lignotubers or stems (epicormic), we compared floral CNglyc content and relative defence of pistillate tissues of resprouters and non-sprouters. The mean whole floret CNglyc content of the three resprouters (tropical *B. celsissima* and temperate *T. speciosissima* and *L. myricoides*) was similar to species that rely on regeneration from seed (Fig 1). Within florets, non-sprouters had significantly higher relative CNglyc content in all pistillate tissues (whole pistils, ovaries, styles and pollen presenters) than resprouters (S4E Fig).

## Discussion

### Comparison of leaf and floral cyanogenic glycosides

At the whole plant level, despite substantial variation in floral CNglyc content between the eleven species studied, floral CNglyc content was higher than foliar in seven species, consistent with our hypothesis based on the ODT [19, 20]. Flowers of several non-proteaceous species similarly contain higher content of CNglycs [14, 16] and other chemical defence metabolites [9, 12] than vegetative tissues, however this pattern is not universal [14, 15, 21] and sampling more individuals across populations would confirm whether these differences are consistent within a species. It is noteworthy that foliar samples in this study were mature and often sclerophyllous leaves, and it would be worthwhile testing soft young leaves which typically have much higher content of CNglycs and other defences [14, 78–80]. Extremely high floral CNglyc content was recorded in several species which had negligible foliar CNglycs (<8 µg CN g$^{-1}$ DW) similar to the pattern identified in *Lomatia* spp. [17]. It is unclear how many species may have chemical defences specific to flowers; however, together with results from several other cyanogenic taxa [31, 32], findings here indicate that species can deploy highly flower-restricted chemical defence.

### Patterns in the distribution of cyanogenic glycosides in floral tissues of Proteaceae

Specificity in floral defence strategies is even more apparent when examining the distribution of CNglycs within flowers which varied markedly between species, with four different and specific patterns identified in which the highest CNglyc content was either restricted to one or two floral tissues, or spread more evenly across multiple floral tissues, especially pistillate tissues. In seven of the eleven taxa, the highest CNglyc content was in one of the pistillate tissues (ovary, style or pollen presenter), consistent with our hypothesis based on optimal defence and apparency theories [6, 19, 20] that valuable tissues retained for fruit/seed formation (ovary) and prominent and vulnerable tissues (style, pollen presenter) would be most defended. These species differentiated into two groups based largely on the specificity of relative CNglyc levels within pistils—one group having the highest levels of CNglycs in ovary or style as well as more even CNglyc distribution across other floral tissues, and the other high relative levels of CNglycs in pollen presenters, the most prominent (i.e. apparent) tissue, and site of secondary pollen presentation, pollination and pollen germination. That only one species (*T. speciosissima*) had negligible CNglyc content in the pollen presenter, but between two and four species had negligible levels of CNglycs in pedicels, ovaries and styles may imply a greater emphasis on defence of the prominent pollen presenter, consistent with Apparency Theory; however, across all species, there was no consistent relationship between pistil length, as a physical measure of vulnerability and apparency, and levels of CNglycs in pistils or pistillate tissues.

Patterns of CNglyc distribution in several species did not conform with our hypothesis that pistillate tissues would be more highly defended; however, the third pattern of high levels of CNglycs in anthers/pollen which grouped *B. celsissima* and *L. myricoides*, is also consistent with ODT and apparency theories, given the fitness value and vulnerability of pollen in species with secondary pollen presentation on the prominent pollen presenter [see also 17]. The specificity of pollen defence was most notable in *B. celsissima* where anther CNglycs accounted for 99.9% of all floral CNglycs, whilst in *L. myricoides* and congeneric species [17], high CNglyc content was in both pollen and loose spiral-walled cells on the pollen presenter, the role of which is unclear, and which are not present in the other species in this study. The defence strategy and grouping of *H. riparia* warrants further investigation, as the high levels of CNglycs

in the perianth of this species, in which anthers were analysed together with perianth, could indicate a greater level of relative pollen defence. The fourth allocation pattern identified in *T. speciosissima* – high CNglyc content and its relatively specific allocation to gynophores and pedicels, the tissue attaching florets and later the developing fruit to the plant – was unexpected and not consistent with our hypothesis. Whether this distribution reflects a defence strategy evolved in response to particular florivores is not known, but relatively high CNglyc content in pedicels has been reported elsewhere [37].

Within-flower variation in chemical defences has previously been reported in other taxa. It is hard to compare the patterns here with those elsewhere as few studies report content in all floral tissues; however, consistent with findings here, several studies report higher defence metabolite content in whole pistils, as predicted by ODT [e.g. 32–34], whereas others do not [e.g. 9, 36, 37]. Even among different almond (*Prunus dulcis*) cultivars, different patterns in floral CNglyc allocation were found–flowers of some cultivars had higher CNglyc content in pistils, and others in petals or sepals [35]. Similarly, defence allocation within pistillate tissues also differed between two *Grevillea* species—whereas the pollen presenters of *G. banksii* had the highest CNglyc content, followed by the ovary and style, *G. bipinnatifida* florets had acyanogenic pollen presenters and ovaries, but released HCN from styles [31]. Both allocations differ to that in *G. robusta* florets, where the greatest CNglyc content was in the gynophore, ovary and style, significantly higher than the pollen presenter. In contrast, a similar pattern within pistils was identified among eight *Lomatia* spp. in which pollen presenters consistently had the highest CNglyc content, despite significant interspecific differences in CNglyc content [17].

Relative levels of investment of CNglycs and other chemical defences in pollen also differ between species, with no consistent pattern [e.g. 81, 82]. Here, relative levels of investment in pollen CNglycs varied widely among the six species analysed, with content ranging from 35 to 9,956 µg CN g$^{-1}$ DW accounting for between 0.1 and 99.9% of the total floral CNglyc pool. While very specific and high relative pollen defence was characteristic of two species, in four species, pollen CNglycs were lower than in other floral tissues, consistent with the small number of reports from other cyanogenic taxa [83–85]. The content of other defence secondary metabolites [e.g. alkaloids [86] and diterpenes [87]] are also generally lower in pollen than in other floral tissues [81, 86], but not always [82, 88–90]. The very specific allocation of CNglycs to pollen in *B. celcissima* and *L. myricoides* [17] is not consistent with the idea that pollen secondary chemistry may be a pleiotropic effect of overflow from other floral tissues [86, 91].

## Relationships between floral defence patterns and other traits

Intra-floral CNglyc allocation patterns could not clearly be explained by variation in other floral traits or factors among species. For example, while the two species with the pollen-dominant CNglyc pattern (*L. myricoides*, *B celsissima*) are white, insect-pollinated species, and *T. speciossisima* with the pedicel-dominant pattern is red and bird-pollinated, overall no consistent pattern with respect to within-flower CNglyc allocations and floral colour was identified. Similarly, interspecific differences in levels of floral CNglycs were not consistently related to differences in flower colour (or likely pollination system)—contrary to our hypothesis and findings of several studies [11, 23, 29, 30], red-flowered species did not have higher levels of CNglycs (S4B Fig). There was no consistent latitudinal pattern in levels of floral CNglycs to support much debated hypotheses around latitudinal patterns in levels of herbivory and chemical defences [58–60, 62, 92]. It is plausible that the significance of these traits for the allocation of floral defences might be revealed by statistical modelling when significantly larger number of species is sampled.

Plant response strategies to disturbances such as fire or cyclones are hypothesised to affect the extent of nutrient allocation to floral organs [93], with a greater allocation of resources to flowers, including chemical defences, predicted in non-sprouting species which rely on regeneration from seed than in resprouting species. Here, whole floret CNglyc levels of non-sprouters and resprouters were similar; however, within florets, CNglyc levels in all pistillate tissues were significantly greater in non-spouter species (S4E Fig). Similarly, flowers of the reseeder *Grevillea banksii* had higher floral (pistillate) CNglyc content than a congeneric resprouter (*G. bipinnatifida*) [31], whereas a study of eight species of *Lomatia* found similar floral defence levels in non-sprouters and resprouters, although the resprouting capacity of *Lomatia* species is not well recorded [17]. Given the larger number of non-sprouter species from tropical rainforests which are not adapted to frequent fire [66], and the small number of resprouter species in this study, it would be worthwhile testing this hypothesis using more species with these different strategies from fire-prone ecosystems [94].

It is well established that both the presence/absence and extent of foliar cyanogenesis can vary between confamilial taxa, and that there can be patterns in their distribution at different taxonomic levels [e.g. 95, 96]. Here both the magnitude of floral cyanogenesis and within-floret distribution patterns varied strongly among the eleven confamilial species, but with no specific taxonomic or phylogenetic pattern with respect to well-supported tribes as an indicator of relatedness [57]. Further investigation of evolutionary and ecological patterns in the diversification of floral cyanogenesis and other traits in more taxa across the Proteaceae phylogeny would be worthwhile [57].

## Roles of cyanogenic glycosides in floral tissues

That confamilial species have such specific and different distributions of potentially costly CNglycs within floral tissues suggests that their allocation is under strong selection, is adaptive and likely evolved in relation to species-specific biotic interactions. Whether these distributions reflect a role in floral defence against fine-scale florivory needs further investigation. Rates of florivory in general are not easily quantified [97] or widely reported [6], and florivore interactions with proteaceous flowers are not documented. However, part-dependent florivory within flowers has been reported in other species [98, 99] and both fine-scale insect damage to florets and large-scale vertebrate florivory were observed (but not quantified) on different study species here and it is plausible that both whole flower and tissue-specific CNglycs play a role in florivore defence [8]. While the lower CNglyc content in some species and floral tissues (e.g. most tissues of *H. riparia* and *H. australasica*) are unlikely to be of ecological significance for defence [100], the extremely high CNglyc content in whole florets and specific floral tissues here are consistent with a defensive function. Some of the tissue-specific contents (8.10–10.05 mg CN $g^{-1}$ DW) are among the highest reported CNglyc levels in floral or vegetative tissues to date, with only tips of sorghum coleoptiles (~25 mg CN $g^{-1}$ DW), pollen of other *Lomatia* spp. (14.6 mg CN $g^{-1}$ DW) and styles of other *M. tetraphylla* samples (21.5 mg CN $g^{-1}$ DW) exceeding this content [17, 18, 101]. A role in defence is further supported by the cyanogenic capacity of floral tissues demonstrating the presence of both a CNglyc and endogenous β-glycosidase [see also 17] which mix upon tissue damage enabling release of volatile HCN [102, 103]. Based on the *dosis letalis* (LD) of HCN for humans (1.4 mg HCN $kg^{-1}$ body weight), dogs (1.5 mg $kg^{-1}$) and rats (~5 mg $kg^{-1}$) [104, 105], the LD for a similarly susceptible insect larva weighing ~50 mg would range from 100–250 ng HCN. Thus, the CNglyc content of just one floret of *G. robusta*, *H. bucculenta*, *M. tetraphylla*, *B. celsissima* and *M. amplexicaulis* exceeds the LD dose by 22–309 times. The high levels of CNglycs in pollen and specificity of localisation to anthers in some species are consistent with a potential role in the deterrence of

florivores [106], non-specialised pollinators and pollen thieves [107–109]. CNglyc content was very high in pollen of several species–*M. tetraphylla* (1186 µg CN g$^{-1}$ DW), *B. celsissima* (8089 µg CN g$^{-1}$ DW) and *L. myricoides* (9956 µg CN g$^{-1}$ DW) and other *Lomatia* spp. [17]. Other toxic secondary metabolites, especially alkaloids, have more often been reported in pollen [e.g. 81, 86, 88, 90]. To date there is no experimental evidence to support potential roles for pollen CNglycs in pollinator attraction [35] or in interactions with microbes on pollen grains [86, 110]. Further, evidence for a role in defence against pollen collection, primarily from studies of bees and rosaceous taxa (*Prunus* spp., *Amygdalus communis*) is inconsistent [83, 85, 111], although the levels in pollen of all species but *T. speciosissima* (34.5 µg CN g$^{-1}$ DW) here are higher than in those other studies (e.g. 108 µg CN g$^{-1}$ DW in *A. communis*; [83]), and the levels at which CNglycs might deter other florivores is unknown [100, 112–115].

While the most recognised function of CNglycs is in defence against herbivores [40, 104], CNglyc allocation in floral tissues may also reflect their potential multiple roles in plant metabolism, for example in seed germination and bud dormancy release [35, 47, 48] and storage, transport and provision of reduced carbon and nitrogen [47, 116–122]. Thus CNglycs in pollen and pollen presenters could play a role in pollen nutrition and germination, as well as defence [123, 124] and CNglycs in ovaries could function as nutrient storage during seed development, and be involved in the regulation of seed germination and provision of nutrients to developing seedlings [e.g. 35, 40, 44, 46, 47, 121, 122]. Cyanogenic diglycosides such as proteacin may be especially important for seed germination and seedling development [46, 120; see below]. More detailed studies of changes in CNglycs during floral and seed development and germination would be required to confirm these roles.

## Specific localisation of cyanogenic mono- and diglycosides in floral tissues

All species in this study were characterised by the same tyrosine-derived CNglycs–the monoglycoside dhurrin and diglycoside proteacin–consistent with all previous studies of Proteaceae [17, 53, 125–127], suggesting biosynthesis of tyrosine-derived CNglycs may be an ancestral trait within the family. Nevertheless, MALDI-MSI identified differences in very fine-scale distributions and relative amounts of the two CNglycs in floral tissues of imaged Proteaceae species. Co-occurring biosynthetically related mono and diglycosides have previously been reported in reproductive tissues of several species including in flowers of proteaceous *Lomatia* and *Hakea* spp. [17, 53], *Prunus dulcis* [35] and *Eucalyptus camphora* [128], as well as in fruit of *P. dulcis* [129]. Fine-scale localisation of different CNglycs in flowers using MALDI-MSI, however, has only been determined in floral buds of *E. cladocalyx* [39], partially open florets of three *Lomatia* species [17], and flower buds of *G. robusta* and *M. tetraphylla* [18].

Consistent with the small number of other studies to report relative content of mono and diglycosides in flowers of other plant families (*Eucalyptus* spp. (Myrtaceae); [39, 128]; *Prunus* spp. (Rosaceae); [35]), MALDI-MSI signal intensities averaged across all floret tissues revealed the monoglycoside dhurrin as the major CNglyc in *H. bucculenta* rachises (93% of CNglycs), similar to the pattern in *L. milnerae* (syn. *L. fraxinifolia*; 95%) [17], and in florets of *G. robusta* (60%). By contrast, proteacin was the major CNglyc in *N. kevedianus* (95%) and *T. speciosissima* (98%) florets—the only species to date in which cyanogenic diglycosides predominate in floral tissues. This is similar to the distribution of CNglycs in foliage across a large number of species, where only in a few species, such as *P. australiana* and *Clerodendrum grayi* do cyanogenic diglycosides comprise a larger proportion of the overall pool of CNglycs [73, 130].

The ratio of the two CNglycs varied significantly between different floral tissues within a species; there were tissues where the two CNglycs co-localised and those where they did not, but there was no consistent pattern across all species. Notably, however, and consistent with

previous reports of ovaries and seeds containing higher proportions of cyanogenic diglycosides [51, 129], proteacin was more abundant than dhurrin in ovules and ovary walls of *T. speciosissima*, *M. tetraphylla*, *N. kevedianus* and *G. robusta*. Although, with the exception of *M. tetraphylla*, relative proteacin content was not necessarily greater in ovules/ovaries than in other floral tissues at the stage of floral development examined. Likewise, proteacin content was highest within *Lomatia* spp. ovules and seeds [17] and *Macadamia integrifolia* young fruit [53]; the diglycosides linustatin and neolinustatin were localised in flax (*Linum usitatissimum*) capsules and seeds [51]; and diglycoside amygdalin was the major cyanogenic constituent in bitter almond (*Prunus dulcis*) cotyledons [129]. That proteacin was present in ovules and seed in all tested Proteaceae taxa is consistent with a possible role for hydrolysis of diglycosides during seed germination and early seedling development [46, 47, 117]. In addition, since changes in relative mono and diglycoside content during floral development in *Prunus* spp. have been implicated in initiating flowering [35], further investigation of tissue-specific regulation of dhurrin and proteacin content during floral and seed development in Proteaceae is warranted.

Differential distributions of dhurrin and proteacin also provided some support for the hypothesis that cyanogenic diglycosides are the more stable form for transport [120, 131] and may be transported within florets and inflorescences, as proteacin co-localised with sucrose in the vasculature of the inflorescence rachis of *H. bucculenta*. Further, in pistils and pedicels of *G. robusta* and *M. tetraphylla*, dhurrin was localised to epidermal layers, consistent with a role in defence, whereas proteacin was more widely distributed throughout internal tissues, including vasculature [see also 18]. By contrast, MALDI-MSI of floral buds of *E. cladocalyx* revealed vascular tissues contained only the monoglycoside prunasin, while the diglycoside amygdalin was restricted to the filaments [39].

The potential transport of CNglycs and the highly tissue-specific localisation of CNglycs within florets of several species raises questions about the site of synthesis of floral CNglycs in Proteaceae, particularly given the highly variable content in leaves. The synthesis of CNglycs in foliage [101, 132–134] and their transport to other tissues has been reported [120, 131]; however, no studies to date have specifically investigated the site of synthesis of floral CNglycs. Studies in *Prunus* spp. have identified CNglyc biosynthesis in seeds [119] and expression of transcripts of CNglyc biosynthetic genes in floral buds [35], providing evidence for CNglyc biosynthesis in non-vegetative tissues in species which also have high levels of foliar CNglycs [135]. Given the very high floral CNglyc content and tissue-specific localisations in the studied Proteaceae species with very low levels of foliar CNglycs (*H. australasica*, *H. riparia*, *L. claudiensis* and *M. tetraphylla*) or essentially acyanogenic foliage (*B. celsissima*, *G. robusta*, *N. kevedianus* and *L. myricoides*), we suggest biosynthesis in floral tissues is likely. The use of MALDI-MSI to detect CNglycs, their biosynthetic intermediates, and structural derivatives [47], in combination with tissue-specific transcript expression studies [e.g. 35, 119] for genes involved in biosynthesis of tyrosine-derived CNglycs [e.g. 136], would confirm whether and which floral tissues are involved in CNglyc synthesis. These analyses would also reveal more about the potential multiple roles of CNglycs in flowers.

## Supporting information

**S1 Table. Collection information and accession numbers for voucher specimens for the eleven Proteaceae species used in this study.** Voucher accession number: MELU—The University of Melbourne Herbarium (Victoria, Australia), NE–NCW Beadle Herbarium, University of New England (New South Wales, Australia).
(PDF)

**S2 Table. The cyanogenic glycoside content of measured as evolved cyanide (μg CN g$^{-1}$ DW) from whole florets and leaves of eleven Proteaceae species (means ± SE, n = 3–5 replicate composite samples from 1–6 plants).** Letters (abc) indicate significant difference in floral or foliar log transformed content between species, using Tukey family grouping test, means that do not share a letter are significantly different.
(PDF)

**S3 Table. The cyanogenic glycoside content of measured as evolved cyanide (μg CN g$^{-1}$ DW) from floral tissues of eleven Proteaceae species (means ± SE, n = 3–5 replicate composite samples from 1–6 plants).** Evolved cyanide content was significantly different in all species ($P \leq 0.001$). Letters (abc) indicate significant difference between floral tissues for the same species, using Tukey family grouping test; means that do not share a letter are significantly different. ND–no data, i.e. for *L. claudiensis* and *N. kevedianus* the "pollen presenter" content is from style and pollen presenter combined, and for *H. australasica*, *H. riparia*, *N. kevedianus*, *L. claudiensis* and *M. amplexicaulis* anthers were analysed with perianth. NA–tissue not present for the species.
(PDF)

**S4 Table. Relative dhurrin and proteacin content of floral and fruit tissues from five Proteaceae taxa and across multiple developmental stages.** The dhurrin to proteacin ratios were derived from MALDI-MSI signal intensities. ND–no data (i.e. tissue was not available in the imaged section). NA–tissue not present/available for the species (i.e. spiral cells are only present in *L. milnerae* (syn *L. fraxinifolia*) and perianth/anthers not available in senescent floral stage). † *L. milnerae* data are from [17]; ‡ *G. robusta* and *M. tetraphylla* young floret data are from [18].
(PDF)

**S1 Fig. Developmental stages of *Grevillea* (Proteaceae) florets showing organisation of floral tissues where pp is pollen presenter, st—style, ov- ovary, ped—pedicel, pe–perianth and an–anthers.** 1 –young, 2 –immature, 3 –partially open, 4 –open, 5 –senescent. Note, gynophore is absent in this species. White scale bar is 5 mm.
(PDF)

**S2 Fig. (A) The variation of tissue-specific floral evolved cyanide content (μg CN g$^{-1}$ DW) and (B) relative evolved cyanide content of six *Grevillea robusta* biological replicates (means ± SE) at partially open stage.** Each point indicates a biological replicate mean of 4 replicates sampled from each individual for each tissue. The mean whole floret content across the six individuals was 3530.18 ± 252.3 μg CN g$^{-1}$ DW. Evolved cyanide content (or relative content) differed significantly between floral tissues ($P \leq 0.0001$). Letters (abc) indicate significant differences between floral tissues, using Tukey HSD family grouping test; means that do not share a letter are significantly different.
(PDF)

**S3 Fig. Representative extracted ion chromatograms (EIC) from LC-MS corresponding to [M + Na]$^{+}$ proteacin (1) at *m/z* 496, and dhurrin (2) at *m/z* 334 in foliar extracts of *P. australiana* and *M. tetraphylla* and whole flower extracts of *G. robusta*, *H. bucculenta*, *N. kevedianus* and *T. speciosissima*.** Mass errors (Δm) were typically ≤ 2 mDa.
(PDF)

**S4 Fig. The relative content of evolved cyanide in specific floral tissues from 11 Proteaceae species grouped by (A) tribe, (B) flower colour, (C) pistil length (mm), (D) climate group, and (E) regeneration strategy.** Bars are means ± SE. Letters (abc) indicate significant

differences at $P < 0.05$ between floral tissues, using Tukey-Kramer *post-hoc* test; means that do not share a letter are significantly different. NA–tissue not available for the group.
(PDF)

## Acknowledgments

We thank Wendy Cooper, Bruce Gray, Rigel Jensen, Diana King, Tom Sayers and Judy and Stephen Williams for floral material; Allison van de Meene for light microscopy images, and Anna Obvintseva for lab assistance. MALDI-MSI was conducted at Metabolomics Australia, The University of Melbourne, an NCRIS initiative under Bioplatforms Australia.

## Author Contributions

**Conceptualization:** Edita Ritmejerytė, Berin A. Boughton, Michael J. Bayly, Rebecca E. Miller.

**Data curation:** Edita Ritmejerytė.

**Formal analysis:** Edita Ritmejerytė, Berin A. Boughton, Rebecca E. Miller.

**Funding acquisition:** Edita Ritmejerytė, Berin A. Boughton, Michael J. Bayly, Rebecca E. Miller.

**Investigation:** Michael J. Bayly.

**Methodology:** Edita Ritmejerytė, Berin A. Boughton, Rebecca E. Miller.

**Supervision:** Berin A. Boughton, Michael J. Bayly, Rebecca E. Miller.

**Validation:** Berin A. Boughton.

**Visualization:** Edita Ritmejerytė.

**Writing – original draft:** Edita Ritmejerytė.

**Writing – review & editing:** Edita Ritmejerytė, Berin A. Boughton, Michael J. Bayly, Rebecca E. Miller.

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
