## [Decision Letter · Decision Letter 0]

24 May 2022

PONE-D-22-02128Interspecific differences in floral chemical defence: diverse organ-specific localisation of cyanogenic glycosides in flowers of ProteaceaePLOS ONE

Dear Dr. Ritmejerytė,

Thank you for submitting your manuscript to PLOS ONE. After careful consideration, we feel that it has merit but does not fully meet PLOS ONE’s publication criteria as it currently stands. Therefore, we invite you to submit a revised version of the manuscript that addresses the points raised during the review process. The two reviewers have provided several suggestions that can improve the manuscript and they point at some important issues that must be dealt with in order for your manuscript to be considered for publication in PLOS ONE.

Both reviewers question how phylogeny has been incorporated in the analyses.

The theoretical framework presented in the introduction is further questioned by reviewer 2. The reference list appears to mostly contain quite old papers on plant apparency and OD theory that do not appear to fit the theme of your work on how selection by pollinators and herbivores may involve both attraction and defence.

The reference to optimal defense/optimal allocation theory is also  questioned by reviewer two relative to the nature of the data presented. The authors should clarify how their data may indeed provide an answer to those theories. 

Technical details related to the chemical determination and quantification are requested by reviewer one and should be added to the manuscript.

The statistical analyses as suggested by reviewer two should be  better explained and sample sizes indicated. It is for example questioned how analyses can be performed with indication of variation when just one tree is sampled per species. Please make sure to not violate the rule of independent data in the analyses.

A suggestion from the academic editor is that as an alternative to hypotheses testing, hypotheses generating statistics could be employed, and the authors could consider to use multivariate statistics to visualize the data. Coloration by any of the detailed traits provided in table 1 in a PCA-plot might for example provide insight into commonalities across Proteaceae and how they allocate cyanogenic glycoside to flower parts.  

We look forward to receiving your revised manuscript.

Kind regards,

Benedicte Riber Albrectsen

Academic Editor

PLOS ONE

**Journal requirements:**

“This work was supported by an Early Career Researcher grant to Rebecca Miller, and the Holsworth Wildlife Research Endowment & the Ecological Society of Australia, the Albert Shimmins Fund Scholarship and Norma Hilda Schuster Scholarship to Edita Ritmejerytė. Edita Ritmejerytė was a recipient of an Australian Postgraduate Award Scholarship. Rebecca Miller’s lectureship received support from the Cybec Foundation.”

“This work was supported by an Early Career Researcher grant to Rebecca Miller, and the Holsworth Wildlife Research Endowment & the Ecological Society of Australia, the Albert Shimmins Fund Scholarship and Norma Hilda Schuster Scholarship to Edita Ritmejerytė. Edita Ritmejerytė was a recipient of an Australian Postgraduate Award Scholarship. Rebecca Miller’s lectureship received support from the Cybec Foundation. The funders had no role in study design, data collection and analysis, decision to publish, or preparation of the manuscript.”

5. We note that [Figure 1] in your submission contain [map/satellite] images which may be copyrighted. All PLOS content is published under the Creative Commons Attribution License (CC BY 4.0), which means that the manuscript, images, and Supporting Information files will be freely available online, and any third party is permitted to access, download, copy, distribute, and use these materials in any way, even commercially, with proper attribution. For these reasons, we cannot publish previously copyrighted maps or satellite images created using proprietary data, such as Google software (Google Maps, Street View, and Earth). For more information, see our copyright guidelines: http://journals.plos.org/plosone/s/licenses-and-copyright.

Natural Earth (public domain): http://www.naturalearthdata.com/.

Reviewers' comments:

Reviewer's Responses to Questions

**Comments to the Author**

1. Is the manuscript technically sound, and do the data support the conclusions?

Reviewer #1: Yes

Reviewer #2: Partly

2. Has the statistical analysis been performed appropriately and rigorously? 

Reviewer #1: Yes

Reviewer #2: I Don't Know

3. Have the authors made all data underlying the findings in their manuscript fully available?

Reviewer #1: Yes

Reviewer #2: Yes

4. Is the manuscript presented in an intelligible fashion and written in standard English?

Reviewer #1: Yes

Reviewer #2: Yes

5. Review Comments to the Author

Reviewer #1: This paper used MALDI-MSI, a relatively new technique, to study cyanogenic glycosides in flowers of 11 Proteaceaes. The results show that allocation patterns were not correlated with other floral traits or phylogenetic relatedness, has rich diversity in floral CNglycs.

1. Line 172-173: There four species only fully open florets were available, which different from other samples. This is equivalent to about a quarter of the samples are of different growth stages.These four species need to be identified. Will it affect the later result analysis and statistics? Please explain.

2. Line 232-233: Please add the name and manufacturer of the instrument and column used in LC-MS analysis.

3. line 371-381：I suggest put this section in Chemical analyses-Quantitative determination of cyanogenesis.

4. Line 514-520: You mention that your results are not consistent with Proteaceae phylogeny.Is it necessary to compare with Proteaceae phylogeny here? Please give a reasonable explanation for the inconsistencies.

5. Line 539-546: The results and discussion in this section are similar to line 518-520 and should be put together. What scientific problems does the phenomenon of no consistent difference in the pattern of CNglyc indicate?

6. Are there any the standards used in the LC–MS? What is the standards? Please write it in detail.

7. Why doesn't fig. 5 show the box plots?

8. Please pay attention to the layout of the picture. There are overlapping labels in Fig.3-7.

Reviewer #2: This is a nice comparative study and the detailed exploration into the distribution of cyanogenic glycosides among flower tissues will make a nice contribution to the body of knowledge around allocation of plant defensive chemistry. My main criticism of the manuscript is that the statistical treatment is rather opaque. I found it difficult to reconstruct exactly what model was being used (test statistics and degrees of freedom would be helpful – given the hierarchical nature of the design, tissues nested within individuals, nested within species, etc., a table seems appropriate).

The introduction really focuses on the chemical ecology of the system (specifically defense allocation, which is nicely laid out), whereas the discussion does into much more depth on the specific aspects of chemistry. Basically, it spends a lot of space discussing things that would not be anticipated in the introduction. Would it be possible to motivate those aspects of the discussion in the introduction?

I think it would be worth explicitly laying out how “apparency” is being used here (line 66). E.g., McCall and Irwin specifically refer to “Floral apparency” as a subset of Optimal Defense Theory. “Apparency Theory” and the reference to Feeny doesn’t seem as appropriate as introduced here – since qualitative vs. quantitative defenses aren’t really being examined here.

L 297: What analysis does this p value pertain to? Is this a correlation between floral and foliar concentrations? If so, show the correlation coefficient. Regardless, I might change the language on line 296 to “failed to detect a relationship” since power is probably not very high with an n of 11 species.

L 288-294, Fig 1. I think the Anova table would be helpful here. It’s difficult to understand exactly what the linear model is here. If these are within species, and n=3-5, how does the model not run out of degrees of freedom (more factor levels than replicates nested within species – is the error being partitioned correctly here?). Is “whole floret” in the model along with the floral parts (which doesn’t seem right)? It seems like “whole foret” vs. “leaf” could boil down to a simple paired t-test (with 10 degrees of freedom), or as a planned contrast between the floral parts lumped together vs. leaf (assuming that the replicates within species are being accounted for). I suggest including the number of replicates within each species on the figure.

L 516: I don’t see a phylogeny (only taxonomy which tells us nothing about relative time since divergence among groups). I think that’s probably fine, since there aren’t enough groups to really do any sort of accounting for phylogeny (though, if available, one could partial out the effect of phylogeny). I do think it is better to refer to this as “taxonomic”, rather than phylogenetic.

6. PLOS authors have the option to publish the peer review history of their article (what does this mean?). If published, this will include your full peer review and any attached files.

Reviewer #1: No

Reviewer #2: No

---

## [Author Response · Author response to Decision Letter 0]

12 Jan 2023

Reviewers' comments:

Comments to the Author

Reviewer #1: This paper used MALDI-MSI, a relatively new technique, to study cyanogenic glycosides in flowers of 11 Proteaceaes. The results show that allocation patterns were not correlated with other floral traits or phylogenetic relatedness, has rich diversity in floral CNglycs.

1. Line 172-173: There four species only fully open florets were available, which different from other samples. This is equivalent to about a quarter of the samples are of different growth stages. These four species need to be identified. Will it affect the later result analysis and statistics? Please explain.

Authors response: The floral developmental stages in this manuscript are irrelevant in terms of statistical comparisons and they have been mentioned only to clarify why anthers, which are embedded within the perianth, could not be removed after dehiscing in fully open floral stage: “Florets of most species were dissected at immature or partially open flower stages prior to anther dehiscence (S1 Fig), but for four species only fully open florets were available (anthers dehisced) thus anthers could not be separated from flowers of these species.” suggests that florets for most of the species analysed were immature or with indehisced anthers, so that pollen would be still contained in the anthers for the ease of separation from perianth. The four species that were dissected at fully open stage have now been listed (L177; “(H. australasica, H. riparia, N. kevedianus and M. amplexicaulis)”). 

Here primary focus was to present distribution patterns of floral CNglycs rather than state differences between floral developmental stages. We do acknowledge in our manuscript that for example H. riparia anthers may contain CNglycs rather than perianth, but we had no floral material available to confirm this, (L611-614; “The defence strategy and grouping of H. riparia warrants further investigation, as the high levels of CNglycs in the perianth of this species, in which anthers were analysed together with perianth, could indicate a greater level of relative pollen defence.”). Consequently, we do not claim statistical differences or significance where the tissues weren’t separated.

The data have been collected over 6 years of different developmental stages across these species based on plant material availability each season (data not shown but presented in Ritmejeryte (2019) PhD thesis), thus we can confirm that the intrafloral distribution patterns of CNglycs are the same across the floral developmental stages. 

2. Line 232-233: Please add the name and manufacturer of the instrument and column used in LC-MS analysis.

Authors response: The name and manufacturer of the instrument and column used in LC-MS analysis have been added as requested (L236-239; “…using an Agilent 1200 HPLC system coupled to an Agilent 6520 Series QTOF-MS (Agilent Technologies, Mulgrave, VIC, Australia) and A Zorbax SB - C18 column (Agilent; 1.8 μm, 2.1 × 50 mm)”).

3. line 371-381：I suggest put this section in Chemical analyses-Quantitative determination of cyanogenesis.

Authors response: We thank the reviewer for a suggestion, however we believe that L371-381 that describes specific CNglycs m/z and retention times is a result and therefore should remain in the results section “Identification of cyanogenic glycosides in Proteaceae flowers” of the manuscript rather than be moved to methods section as suggested by the reviewer. 

4. Line 514-520: You mention that your results are not consistent with Proteaceae phylogeny.Is it necessary to compare with Proteaceae phylogeny here? Please give a reasonable explanation for the inconsistencies.

Authors response: this was the first study investigating the allocation of floral defences at such fine scale across multiple genera. While we hypothesised that the defence allocation patterns could be more similar between closely related taxa than those less related. This is not necessarily surprising, as for example Lamont (1) found dissimilarities in CNglyc presence of different floral tissues in two congeneric Grevillea species, which have been hypothesised to be driven by their different adaptations to fire. Therefore, here we discuss phylogeny or species relatedness as one of the possible factors that could drive the allocations of floral defence, but we also discuss floral colour, putative pollinators and floral structure as other potential traits driving floral defences. We have also changed the term “phylogenetic” to “taxonomic” relatedness throughout the manuscript and have changed the species order in table 1 from alphabetical order to taxonomic order.

5. Line 539-546: The results and discussion in this section are similar to line 518-520 and should be put together. What scientific problems does the phenomenon of no consistent difference in the pattern of CNglyc indicate?

Authors response: We thank the reviewer for a suggestion, however due to the two sections discussing different theories, such as species relatedness, climate adaptations and adaptations to respond to abiotic disturbances such as fire or cyclone, we have decided to keep the results regarding species phylogenetic relatedness, and the section about latitude and disturbance response strategies separate. 

We acknowledge the limitations of this study (e.g. L673-676; “Given the larger number of non-sprouter species from tropical rainforests which are not adapted to frequent fire [66], and the small number of resprouter species in this study, it would be worthwhile testing this hypothesis using more species with these different strategies from fire-prone ecosystems [95].”), that more species should be sampled to test these theories in order to discover any patterns. We have added the following sentence to further indicate the limitations of our work and directions for future studies (L659-662; “It is plausible that the significance of these traits towards the allocation of floral defences might be revealed by statistical modelling when significantly larger number of species is sampled.”)

In following studies, multiple plants of each species, and many more species should be collected, dissected and analysed to fill in the gaps of this first study. Statistical modelling could then be applied to provide more insight of which (or perhaps all) of these traits play a role in floral defence allocations. Moreover, research on pollinators, florivores and other flower visitors for these species are needed to confirm pollinator systems and the florivore pressures. However, dissecting florets at this fine scale is labour intensive, plant material for multiple plants is rarely available, the species are distributed across Australia (and the rest of Southern Hemisphere), with very short flowering seasons, each of which complicate feasibility of this work by one research group. 

6. Are there any the standards used in the LC–MS? What is the standards? Please write it in detail.

Authors response: For this work, the foliar extract of P. australiana was used as the LC-MS standard because it contains both CNglycs, dhurrin and proteacin [2] as described in the experimental section with wording slightly changed (L229-232; “Because Proteaceae are known to contain tyrosine-derived CNglycs [53], floral extracts of the six species were analysed alongside an extract from P. australiana foliage as a standard because it is known to contain two tyrosine-derived CNglycs, dhurrin and proteacin [73]”) and result section (L365-366; “…and a Polyscias australiana leaf MeOH extract known to contain both dhurrin and proteacin [73] as a standard”). It was confirmed using TLC that these species extracts had only two cyanogenic regions, i.e. two CNglycs which co-eluted with cyanogenic compounds from P. australiana as further shown in [3].

7. Why doesn't fig. 5 show the box plots?

Authors response: box plots in Figs 3, 4, 6 and 7 show the ratios of two CNglycs within specific floral tissues measured as signal intensity using MALDI, whereas Fig. 5, visualises the localisation of CNglycs in rachis, in which the visual distinction between tissue types is not very clear for a comparison to be plotted in box plot due to the small section size, but we believe this image is important to show the localisation of cyanogenic diglycosides in vascular tissues, while the monoglycoside dhurrin was localised in epidermis – both consistent with their proposed functions.

8. Please pay attention to the layout of the picture. There are overlapping labels in Fig.3-7.

Authors response: Thank you for pointing this out. We believe that the labels of the figures moved when converting the submitted documents from PPT to PDF, and this has been fixed as shown below, in fig. 5 as an example: 

 

Reviewer #2: This is a nice comparative study and the detailed exploration into the distribution of cyanogenic glycosides among flower tissues will make a nice contribution to the body of knowledge around allocation of plant defensive chemistry. My main criticism of the manuscript is that the statistical treatment is rather opaque. I found it difficult to reconstruct exactly what model was being used (test statistics and degrees of freedom would be helpful – given the hierarchical nature of the design, tissues nested within individuals, nested within species, etc., a table seems appropriate).

Authors response: due to limited sample size we were unable to use statistical modelling for this work, except for Supplementary Fig. 4, where all species were pooled to look into interspecific patterns. In Fig. 1 Tukey HSD post hoc were performed separately for each species, with statistical significance indicated by different letters (L153-155; “Letters (abc) indicate significant differences between floral tissues for the same species, using Tukey HSD family grouping test; means that do not share a letter are significantly different.”). Finally, for Fig. 2, we initially attempted PCA analyses, however due to the nature of the dataset, where for example anther CNglyc concentrations could not be quantified for some species, and concentrations of other tissues (e.g. gynophores or pedicels) for some species are unavailable, we are unable to perform PCA. We therefore chose to visually present the mean relative concentrations for each species (Fig. 2) and how they would group, but we do not draw conclusions indicating statistical significance, or the strength of these groups.

The introduction really focuses on the chemical ecology of the system (specifically defense allocation, which is nicely laid out), whereas the discussion does into much more depth on the specific aspects of chemistry. Basically, it spends a lot of space discussing things that would not be anticipated in the introduction. Would it be possible to motivate those aspects of the discussion in the introduction?

Authors response: We have now included more details (in red) on the chemistry of the cyanogenic glycoside structures in the introduction as requested by the reviewer (L94-99; “This is the case with CNglycs, a group of nitrogen (N)-based secondary metabolites found in over 3,000 plant species [40] with currently 112 structures known [41], they vary in the structures of their amino acid precursor, aglycones and the type and the number of sugar moieties The primary role of CNglycs is herbivore deterrence by releasing toxic hydrogen cyanide (HCN) upon tissue disruption [42, 43], particularly characteristic of monoglucosides (those containing one sugar moiety). However, CNglycs, especially those containing more than one sugar moiety (e.g. diglycosides) may play additional roles in plant metabolism.”).

I think it would be worth explicitly laying out how “apparency” is being used here (line 66). E.g., McCall and Irwin specifically refer to “Floral apparency” as a subset of Optimal Defense Theory. “Apparency Theory” and the reference to Feeny doesn’t seem as appropriate as introduced here – since qualitative vs. quantitative defenses aren’t really being examined here.

Authors response: In our manuscript, Feeny (1976) [4] was cited as this work was the first to define the term “apparency” in an ecological context, although not specific to floral defences. Here we used floral apparency theory as a subset to optimal defence theory McCall and Irwin (2010), as noted by the reviewer and as cited in our manuscript (L66-68) “The ODT and related Apparency Theory [6, 22] also predict that more conspicuous or accessible flowers or their specific tissues will be more defended.”. The ODT and apparency theory are further described in the following two paragraphs in introduction, firstly in relation to floral colour where red flowers are likely to be more defended than white; and secondly, in relation to the cost and apparency of different floral tissues, such as ovaries being the most costly and they are retained for seed formation, and pistillate tissues being most conspicuous.

L 297: What analysis does this p value pertain to? Is this a correlation between floral and foliar concentrations? If so, show the correlation coefficient. Regardless, I might change the language on line 296 to “failed to detect a relationship” since power is probably not very high with an n of 11 species.

Authors response: We have changed the wording as suggested by the reviewer (in red) and added the model and correlation coefficient. The sentence now reads: “Foliar CNglyc concentrations also differed significantly between species (0.1-1,644 μg CN g-1 DW), and generalized linear model failed to detect a relationship between floral and foliar CNglyc concentrations across species (log(whole flower CNglyc concentration)~log(foliar CNglyc concentration)+(1|species), R2=0.02, P=0.29)” (L299-302). 

L 288-294, Fig 1. I think the Anova table would be helpful here. It’s difficult to understand exactly what the linear model is here. If these are within species, and n=3-5, how does the model not run out of degrees of freedom (more factor levels than replicates nested within species – is the error being partitioned correctly here?). Is “whole floret” in the model along with the floral parts (which doesn’t seem right)? It seems like “whole foret” vs. “leaf” could boil down to a simple paired t-test (with 10 degrees of freedom), or as a planned contrast between the floral parts lumped together vs. leaf (assuming that the replicates within species are being accounted for). I suggest including the number of replicates within each species on the figure.

Authors response: Statistical analysis presented in Fig. 1 were ANOVA and Tukey HSD post hoc performed separately for each species, with statistical significance indicated by different letters, rather than a model (L153-155; “Letters (abc) indicate significant differences between floral tissues for the same species, using Tukey HSD family grouping test; means that do not share a letter are significantly different.” and L273-275; “Transformed data were compared using Tukey HSD post-hoc test with significance at P ≤ 0.05 using Minitab (version 17)”). We do agree that whole floret and leaf concentrations could be compared separately, but in these comparisons presented here whole flower and foliar concentrations have been included on purpose, to present how much whole flower or foliar CNglyc concentrations (most commonly reported in research articles) differ from tissue-specific concentrations, often by degrees of magnitude.

L 516: I don’t see a phylogeny (only taxonomy which tells us nothing about relative time since divergence among groups). I think that’s probably fine, since there aren’t enough groups to really do any sort of accounting for phylogeny (though, if available, one could partial out the effect of phylogeny). I do think it is better to refer to this as “taxonomic”, rather than phylogenetic.

Authors response: thank you for pointing this out. We have now changed the wording to “taxonomic” rather than “phylogenetic” relationships. We have also re-arranged the order of the species in table 1 from alphabetical, to taxonomic to indicate taxonomic relationships of the 11 species discussed in our study.

 References:

1. Lamont BB. Injury-induced cyanogenesis in vegetative and reproductive parts of two Grevillea species and their F1 hybrid. Annals of Botany. 1993;71(6):537-42. doi: 10.1006/anbo.1993.1069.

2. Miller RE, Tuck KL. The rare cyanogen proteacin, and dhurrin, from foliage of Polyscias australiana, a tropical Araliaceae. Phytochemistry. 2013;93:210-5. Epub 2013/04/10. doi: 10.1016/j.phytochem.2013.03.004. PubMed PMID: 23566716.

3. Ritmejerytė E, Miller RE, Bayly MJ, Boughton BA. Chapter 3 - Visualization of cyanogenic glycosides in floral tissues. In: Beale DJ, Hillyer KE, Warden AC, Jones OAH, editors. Applied Environmental Metabolomics: Academic Press; 2022. p. 29-44.

4. Feeny P. Plant apparency and chemical defense. In: Wallace JW, Mansell RL, editors. Recent Advances in Phytochemistry. Biochemical interaction between plants and insects. 10: Springer; 1976. p. 1-40.

5. McCall AC, Fordyce JA. Can optimal defence theory be used to predict the distribution of plant chemical defences? Journal of Ecology. 2010;98(5):985-92. doi: 10.1111/j.1365-2745.2010.01693.x.

---

## [Decision Letter · Decision Letter 1]

27 Mar 2023

PONE-D-22-02128R1Interspecific differences in floral chemical defence: diverse organ-specific localisation of cyanogenic glycosides in flowers of ProteaceaePLOS ONE

Dear Dr. Ritmejerytė,

Thank you for submitting your manuscript to PLOS ONE. After careful consideration, we feel that it has merit but does not fully meet PLOS ONE’s publication criteria as it currently stands. Therefore, we invite you to submit a revised version of the manuscript that addresses the points raised during the review process.

Although reviewers' comments are carefully addresses, several minor editorial corrections should be made before proceeding.

The first is the use of the term "concentration(s)". Since cyanogenic glycosides are expressed as μg per g tissue DW but not in a certain volume, "concentration(s)" should be substituted by "content". Please see e.g., 10.11613/BM.2013.017 for further information.

Please avoid using vernacular phrases such as e.g., "florets of 11 Proteaceae" or "in flowers of Proteaceae" but use scientifically acceptable terminology and do not spare words to describe an object of the study, for example "florets of 11 studied species from (belonging to) the Proteaceae family" or "in flowers of (studied) species from the Proteaceae family".

The main title might be misleading and a reader might conclude the authors report a review article of most recent literature regarding the content of cyanogenic glycosides across more than 80 genera of the Proteaceae family. Please specify here how many species you studied, for example: "...in flowers of 11 species from the family Proteaceae".

Please also consider a minor remark by Reviewer #2 related to L300.

We look forward to receiving your revised manuscript.

Kind regards,

Branislav T. Šiler, Ph.D.

Academic Editor

PLOS ONE

Journal Requirements:

Reviewers' comments:

Reviewer's Responses to Questions

**Comments to the Author**

1. If the authors have adequately addressed your comments raised in a previous round of review and you feel that this manuscript is now acceptable for publication, you may indicate that here to bypass the “Comments to the Author” section, enter your conflict of interest statement in the “Confidential to Editor” section, and submit your "Accept" recommendation.

Reviewer #1: All comments have been addressed

Reviewer #2: All comments have been addressed

2. Is the manuscript technically sound, and do the data support the conclusions?

Reviewer #1: Yes

Reviewer #2: Yes

3. Has the statistical analysis been performed appropriately and rigorously? 

Reviewer #1: Yes

Reviewer #2: Yes

4. Have the authors made all data underlying the findings in their manuscript fully available?

Reviewer #1: Yes

Reviewer #2: Yes

5. Is the manuscript presented in an intelligible fashion and written in standard English?

Reviewer #1: Yes

Reviewer #2: Yes

6. Review Comments to the Author

Reviewer #1: (No Response)

Reviewer #2: One minor thing: ref Line 300: Was this a general linear model, or generalized linear model? I think general linear mixed model?

7. PLOS authors have the option to publish the peer review history of their article (what does this mean?). If published, this will include your full peer review and any attached files.

Reviewer #1: No

Reviewer #2: No

---

## [Author Response · Author response to Decision Letter 1]

29 Mar 2023

Reviewer #2: One minor thing: ref Line 300: Was this a general linear model, or generalized linear model? I think general linear mixed model?

Authors response: we thank the reviewer for noting this, the text has been corrected to “general linear model”.

---

## [Editor Report · Decision Letter 2]

30 Mar 2023

PONE-D-22-02128R2Diverse organ-specific localisation of a chemical defence, cyanogenic glycosides, in flowers of eleven species of ProteaceaePLOS ONE

Dear Dr. Ritmejerytė,

Thank you for submitting your manuscript to PLOS ONE. After careful consideration, we feel that it has merit but does not fully meet PLOS ONE’s publication criteria as it currently stands. Therefore, we invite you to submit a revised version of the manuscript that addresses the points raised during the review process. The authors have failed to address the academic editor's comment from the previous review round: The first [problem] is the use of the term "concentration(s)". Since cyanogenic glycosides are expressed as μg per g tissue DW but not in a certain volume, "concentration(s)" should be substituted by "content". Please see e.g., 10.11613/BM.2013.017 for further information. No respective changes were made in the text and no rebuttal was provided regarding this concern.

We look forward to receiving your revised manuscript.

Kind regards,

Branislav T. Šiler, Ph.D.

Academic Editor

PLOS ONE
---

## [Author Response · Author response to Decision Letter 2]

13 Apr 2023

Please also consider a minor remark by Reviewer #2 related to L300.

Authors response: we thank the reviewer for noting this, the text has been corrected to “general linear model”.

---

## [Editor Report · Decision Letter 3]

14 Apr 2023

Diverse organ-specific localisation of a chemical defence, cyanogenic glycosides, in flowers of eleven species of Proteaceae

PONE-D-22-02128R3

Dear Dr. Ritmejerytė,

We’re pleased to inform you that your manuscript has been judged scientifically suitable for publication and will be formally accepted for publication once it meets all outstanding technical requirements.

Kind regards,

Branislav T. Šiler, Ph.D.

Academic Editor

PLOS ONE
---

## [Editor Report · Acceptance letter]

19 Apr 2023

PONE-D-22-02128R3 

Diverse organ-specific localisation of a chemical defence, cyanogenic glycosides, in flowers of eleven species of Proteaceae 

Dear Dr. Ritmejerytė:

I'm pleased to inform you that your manuscript has been deemed suitable for publication in PLOS ONE. Congratulations! Your manuscript is now with our production department. 

Kind regards, 

on behalf of

Dr. Branislav T. Šiler 

Academic Editor

PLOS ONE